# Design Principles Governing the Development of Theranostic Anticancer Agents and Their Nanoformulations with Photoacoustic Properties

**DOI:** 10.3390/pharmaceutics14020362

**Published:** 2022-02-04

**Authors:** Stavroula G. Kyrkou, Eirinaios I. Vrettos, Dimitris Gorpas, Timothy Crook, Nelofer Syed, Andreas G. Tzakos

**Affiliations:** 1Department of Chemistry, Section of Organic Chemistry and Biochemistry, University of Ioannina, 45110 Ioannina, Greece; stavroylakyrkoy@gmail.com (S.G.K.); renos.vrettos@gmail.com (E.I.V.); 2Institute of Biological and Medical Imaging, Helmholtz Zentrum München, Neuherberg, D-85764 Oberschleißheim, Germany; dimitrios.gkorpas@helmholtz-muenchen.de; 3Chair of Biological Imaging, Technische Universität München, D-81675 Munich, Germany; 4John Fulcher Neuro-Oncology Laboratory, Department of Brain Sciences, Division of Neuroscience, Faculty of Medicine, Imperial College London, London W12 0NN, UK; 5Institute of Materials Science and Computing, University Research Center of Ioannina (URCI), 45110 Ioannina, Greece

**Keywords:** cancer, theranostics, photoacoustic, phototherapy, nanoformulations

## Abstract

The unmet need to develop novel approaches for cancer diagnosis and treatment has led to the evolution of theranostic agents, which usually include, in addition to the anticancer drug, an imaging agent based mostly on fluorescent agents. Over the past few years, a non-invasive photoacoustic imaging modality has been effectively integrated into theranostic agents. Herein, we shed light on the design principles governing the development of theranostic agents with photoacoustic properties, which can be formulated into nanocarriers to enhance their potency. Specifically, we provide an extensive analysis of their individual constituents including the imaging dyes, drugs, linkers, targeting moieties, and their formulation into nanocarriers. Along these lines, we present numerous relevant paradigms. Finally, we discuss the clinical relevance of the specific strategy, as also the limitations and future perspectives, and through this review, we envisage paving the way for the development of theranostic agents endowed with photoacoustic properties as effective anticancer medicines.

## 1. Introduction

Cancer has been known to mankind for centuries as a major fatal disease, causing millions of deaths worldwide annually [1]. It has been an unmet challenge for the scientific community to comprehend its complex nature and, consequently, to establish methods for its detection and effective management. Although remarkable and innovative strategies have been developed, the intrinsic characteristics of cancer including metastasis, heterogeneity, and adaptability leading to therapy resistance remain formidable obstacles to successful management [2].

Fundamental to the effective management of cancer is the early diagnosis and clinical detection. Multiple diagnostic modalities, such as magnetic resonance imaging (MRI), positron emission tomography (PET), single-photon emission computed tomography (SPECT), computerized tomography (CT), X-ray, and ultrasound (US) are established in clinical practice [3]. However, the need to develop innovative low-cost and time-effective techniques with superior optical imaging properties and penetration depth remains urgent [4,5]. The requirement to further understand the pathogenesis of cancer has motivated researchers to develop less harmful and simultaneously more efficient in vivo diagnostic tools, including fluorescence and photoacoustic imaging probes with high penetration efficiency and enhanced bioimaging properties [6]. Photoacoustic imaging (PAI) constitutes an emerging approach that possesses the aforementioned desired properties and can be utilized as an efficient molecular detection method [7]. Of course, the critical intervention after the early detection of malignant disease is an effective treatment. Standard treatment options typically involve surgery, followed by radiation and/or chemotherapy. Nevertheless, chemotherapeutic drugs are often limited in their efficacy and have adverse side effects mainly due to their poor pharmacokinetic properties, their non-specific distribution throughout the body causing inevitable peripheral toxicity, and acquired resistance of cancer cells [8]. The availability of new analytical techniques has led to a better understanding of the mechanistic basis of tumor adaptation and has facilitated the development of new conjugates that selectively target tumors, diminishing the uncontrolled toxicity [8]. Specifically, such conjugates usually consist of molecules that are selectively recognized by specific receptors that are uniquely expressed or overexpressed on the surface of malignant tumor cells and lead to the internalization of the entire conjugate (e.g., peptides [9]). Thus, the drug can be released or activated primarily by the tumor microenvironment and selectively act cytotoxically within the tumor area [10]. A non-invasive treatment strategy that has been extensively utilized to treat cancer is phototherapy. Phototherapy is based on the usage of different wavelengths of the electromagnetic spectrum and involves photodynamic therapy (PDT) and photothermal therapy (PTT). PDT combines a photosensitizing agent or photosensitizer, that is designed to eliminate abnormal cells by generating singlet oxygen (^1^O_2_), along with a light source (often a laser) that triggers the drug activation (excitation) [11]. The utilized photosensitizer remains non-toxic to tissues until it is activated on demand by the light source. On the other hand, PTT does not depend on the generation of radical oxygen species but involves the conversion of the absorbed light to heat to destroy vicinal diseased cells. Similar to PDT, this approach can selectively eliminate tumors and spare healthy tissues. Both therapeutic modalities are often utilized in combination with PAI and thus, they will be extensively discussed in Section 4.2 and Section 4.3.

The combination of the aforementioned characteristics of diagnostic and therapeutic modalities has led to the development of theranostic molecules, which have emerged as alternatives to traditional therapeutic and diagnostic approaches and are likely to be major contributors to personalized medicine. The term “theranostic” (therapy + diagnosis) was first mentioned by John Funkhouser in 1998 and describes an agent with simultaneous therapeutic and diagnostic properties [12]. A theranostic agent usually consists of an imaging agent coupled to a therapeutic agent through a spacer or a linker. In certain circumstances, a single agent has both imaging and therapeutic properties. Such a representative example is indocyanine green (ICG), a small molecular dye that is endowed with fluorescence, photoacoustic properties, and photoconversion ability (therapy), as shown by Zhong et al. [13,14]. In addition, its structure allows the installation of a targeting moiety to render it as a targeted theranostic agent with superior properties [15]. It is worth mentioning that a theranostic agent can be organized into more complex structures (e.g., nanostructures) [16], or encapsulated within structures (e.g., microparticles and nanoparticles) [17] usually to enhance its photophysical properties and pharmacokinetic profile, and, thus, to improve its absorption, distribution, metabolism, and excretion (ADME) properties [18].

This review focuses mainly on the principles governing the design of theranostic agents that exert their diagnostic functions through photoacoustic imaging. An extensive analysis of the photoacoustic imaging theory and the dyes that can be utilized for this purpose are displayed (the common dyes are summarized in Figure 1a), and along these lines, we present multiple representative examples of theranostic agents. These agents comprise small organic dyes suitable for photoacoustic imaging, tethered via various linkers (cleavable, non-cleavable or self-immolative) to different anticancer payloads to produce “classic” theranostics, consisting of the drug-linker-bioimaging agent motif (two representative examples are presented in Figure 1b). Moreover, the installation of various tumor-homing elements (e.g., peptides) might further enhance anti-cancer efficacy (Figure 1c). We pay particular attention to theranostic agents with photoacoustic properties whose therapeutic function is accomplished via photodynamic or photothermal therapy (two examples are presented in Figure 1d). Several paradigms also involve targeting moieties and assembly into complex structures, such as nanoparticles (NPs), micelles, and microbubbles. Finally, we reference specific clinical studies, which demonstrate the potential pre-clinical translation of such agents into clinical settings, illustrating the promise of such approaches in cancer medicine, and we also state the limitations and future perspectives.

## 2. Photoacoustic Imaging

Molecular imaging is a widespread technique exploited to visualize and study biological systems at a molecular and cellular level, using either ionizing or non-ionizing radiation, and it has found wide applicability in preclinical and clinical settings in oncology. When light meets a tissue, various interactions are probable, including light scattering or absorption with an inverse relationship between depth of penetration and spatial resolution [19]. Fluorescence is a remarkable imaging technique that is routinely used for monitoring complex living systems. However, the high scattering ability of light degrades the consequent spatial resolution at increased depths, limiting its applicability to superficial levels. The depth of penetration has been progressively increased by the development of new near-infrared (NIR) dyes (700–950 nm called NIR-I dyes & 1000–1700 nm NIR-II dyes) that act within longer wavelengths. Further improvement is still required regarding NIR dyes and the development of non-invasive techniques, and along these lines, an emerging technology, known as photoacoustic imaging, has shown considerable promise.

Photoacoustic imaging is a hybrid non-ionizing technique based on the photoacoustic effect and refers to the interaction of electromagnetic radiation (which includes optical, radio frequency waves and microwaves) intending to generate acoustic waves. The photoacoustic effect was first described by Alexander Graham Bell in 1880, who noticed that an object illuminated by intermittent sunlight could produce acoustic waves [20]. Thus, a nanosecond-pulsed laser can be used to irradiate a biological tissue (optical excitation), and then a US array transducer can be used for detection of the signal (ultrasonic detection) [21].

The processes occurring within the tissue which lead to the output signal can be described in a few steps:A short-pulsed laser, as a source of energy, transfers photons to a target tissue.The energy of the photons is absorbed by endogenous or exogenous contrast agents with optical properties causing activation.Τhe absorbed optical energy is partially or completely converted into heat.Τhe thermoelastic tissue heats up and expands.Τhe heat is transferred to cooler areas and the tissue contracts.Τhe expansion and contraction of the tissue cause pulsatile pressure changes.These, in turn, lead to the production of acoustic pressure waves.Broadband acoustic waves are detected by a US transducer and processed for image generation [22,23,24] (Figure 2).

A typical PAI system consists of major components including a short-pulsed laser; a system for ultrasonic transduction; a data acquisitor; a system for imaging representation and digitization; and, finally, a computer for data combination and shaping of the image [22]. An efficient and functional technique using the photoacoustic phenomenon is photoacoustic tomography (PAT). In the last decade, PAT has been used in a wide range of applications [25] and multiple clinical trials [26]. Further, PAT has three technical applications: photoacoustic microscopy (PAM), photoacoustic computed tomography (PACT), and photoacoustic endoscopy (PAE) [25].

Notably, Bozhko et al. [27] utilized the photoacoustic modality along with intravascular NIRF (near-infrared fluorescence) and IVUS (intravascular ultrasound) to develop a tri-modal NIRF-IVUS-optoacoustic catheter, which was validated in phantoms and excised rabbit aortas. This novel catheter promoted the exact determination of the variations of blood absorption at various hematocrit values and resulted in the accurate quantification of the observed NIRF signal. This novel approach could be further utilized to study plaque pathogenesis and assess their potential to rupture; also it could be applicable to various catheter-based clinical procedures.

These techniques share several advantages over other classic imaging techniques, such as fluorescence [7]. Specifically, the detection of acoustic rather than optical signals results in lower signal scattering and better visual separation, i.e., biological structures from organelles to organs can be visualized at multiscale; it can provide a range of longer wavelengths into the microwave band (300 MHz–3 GHz) and achieve deeper penetration; for optical excitation, wavelengths in the visible and near-infrared part of the spectrum are used; specific inherent tissue chromophores such as hemoglobin, melanin [28], DNA/RNA [29], and water or lipids [30] could be exploited for acoustic imaging. These techniques also offer low background absorption and m not require the use of exogenous agents in certain cases. In addition, the temperature and the pressure increase are below 0.1 K and 10 Pa, respectively, thus avoiding the destruction of normal tissues. Additional advantages, such as the use of non-ionizing radiation, sensitivity, clinically relevant depths (up to several cm penetration depending on the utilized technique [31]), and real-time applications, render these techniques promising for the detection of multiple diseases, including cancer [21,25,32,33].

### 2.1. Basic Principles of Photoacoustic Contrast Agents

Endogenous chromophores, such as tissue components, can be used on a large scale without damaging tissues [34], but their use is limited since they are not amenable to modifications. Thus, the necessity to create modifiable exogenous contrast agents is of obvious importance. Along these lines, small-molecule dyes have come to the forefront, mostly due to their biocompatibility and their immediate and complete clearance from the body (~1 nm in size), which prevents unwanted accumulation and consequent toxicity [32].

For a small molecular dye to possess photoacoustic properties, it should satisfy certain principles. The most crucial factor is the existence of a series of continuous π-bonds and polyaromatic ring structures. These allow their Highest Occupied Molecular Orbital (HOMO) and their Lowest Unoccupied Molecular Orbital (LUMO) to approach each other, to achieve the maximum possible absorption of the offered photons to excite the electrons of the probes. The structure is often complemented with electron-donating (e.g., -OH, -NH_2_) or electron-withdrawing groups (e.g., -NO_2_, -C≡N) at different positions to adjust the spectral range. In addition, hydrophilic groups, triplet state quenchers, or stabilizing groups are often invoked to optimize the physicochemical properties (e.g., solubility, molar extinction coefficient, and photo-bleaching) [35]. Although the exact mechanisms underlying the existence of photoacoustic properties remain partially obscure, the majority of photoacoustic agents share certain characteristics:A high molar extinction coefficient, to maximize the probability of an electronic transition.A sharply peaked, characteristic absorption spectrum, to ensure that it will not overlap with other peaks by spectral mixing even at low molecular concentrations.Peak absorption in the optical window: 600–1100 nm [4], to limit the amount of radiation absorbed by endogenous chromophores and, thus, optimize the depth of penetration.High photostability, to ensure that light irradiation does not alter their spectral features.Low quantum yield, so most of the non-radiant energy can be converted to heat.Small size (<2 nm) to avoid accumulation within tissues.A high number of rotating bonds which is closely correlated to the intramolecular charge transfer phenomenon [7].

Imaging dyes are often accommodated with tumor-targeting ligands or encapsulated into nanocarriers to enhance their half-life and photoacoustic signal [19,35,36,37,38].

Contrast agents that are based on a low-molecular weight organic dye as a core structure and satisfy the above characteristics have been exploited for a wide range of PAI applications. Representative examples involve structures based on porphyrin, semicyanine, heptamethine cyanine, perylene-diimide, aza-BODIPY, benzobisthiadiazole [36,39], methylene blue [40,41,42], Alexa Fluor [43], and Evans Blue [44] (Figure 3). Remarkable efforts have already been made to summarize and categorize the photoacoustic probes [6,7,36,45,46,47,48,49,50,51] and their synthetic strategies [19,52,53,54]. However, this review is not limited solely to imaging applications, but focuses mainly on compounds bearing simultaneous photoacoustic imaging and treatment capability.

## 3. Design Strategies for Cancer Theranostics with Photoacoustic Properties

Theranostic molecules endowed with PAI properties used for cancer diagnosis and treatment typically consist of a PA probe and a distinct cytotoxic payload. In certain cases, the imaging agent may also possess inherent cytotoxicity and the use of an additional anticancer drug can be avoided [55]. Adverse effects typically associated with cytotoxic chemotherapy are well recognized and include nausea and vomiting, myelosuppression, gastrointestinal toxicity, peripheral neuropathy, and alopecia [56]. These effects are predictable, given that a non-targeted toxic agent is introduced into the body. To increase the selective delivery of anticancer agents to tumors and minimize exposure to healthy tissues, tumor-homing elements, such as small molecules, peptides, proteins, and antibodies, can be utilized (targeted therapy). Targeted therapy, at least in part, exploits the microenvironmental differences between cancer cells and normal cells, which differ greatly in their metabolism [57,58]. For instance, they differ in the way they sense oxygen [59,60], they have different signaling pathways [61], and they present variations in their expression levels of certain receptors and enzymes [62,63,64]. Theranostic agents can be composed of various linkers tethering the individual components conferring the required properties, for example, water solubility (i.e., sulfonate groups), bioavailability (i.e., polyethylene glycol (PEG) groups), and stability in biofluids, such as blood (i.e., non-cleavable bonds). Linkers play a crucial role within bioconjugates and their manipulation might interfere (positively or negatively) with the overall bioactivity profile of the conjugate [18,65,66,67,68,69]. Finally, molecules can be encapsulated or organized into micelles, nanoparticles, and other complex nanoformulations [70]. The generalized architecture to develop a theranostic endowed with photoacoustic properties is presented in Figure 4. Its partial compartments can be chosen accordingly to fulfill the desired requirements of every unique patient (personalized medicine). Therefore, the proper assembly of a potent theranostic agent with photoacoustic properties requires certain principles to be followed, and they will be analyzed below.

### 3.1. Selecting the Optimal Imaging Agent during the Design of Cancer Theranostics with Photoacoustic Properties

When constructing a new theranostic agent to possess photoacoustic properties, the optimal dye should be chosen. If the dye is synthesized de novo, it should be developed based on the principles described in Section 2.1. Otherwise, a dye among the representative examples presented in Figure 3 can be selected according to the requirement for characteristic groups (e.g., free -COOH or -NH_2_) or specific photophysical properties (e.g., emission wavelength). For instance, if the utilized anticancer drug possesses a free -NH_2_ group (e.g., gemcitabine), a dye with an electrophilic carbon can be selected (e.g., heptamethine cyanine). Most examples in the current literature utilize cyanine- and BODIPY-based core structures for the development of theranostics with photoacoustic properties. The most representative examples of chromophores utilized within cancer theranostics are analyzed below and are also summarized in Table 1.

Cyanine belongs to the family of polymethine dyes. These are small organic molecules whose structure consists of the following basic characteristics: a polymethine bridge joining two heterocycles containing aromatic nitrogen, such as pyrrole, pyridine, indole, thiazole, and benzothiazole etc., linked by π-conjugation. The extension of the polymethine chain by a vinylene moiety leads to bathochromic shift (approximately 100 nm per CH=CH) [13,47]. Thus, pentamethine (Cy5) and heptamethine cyanine (Cy7) derivatives may show absorption in NIR (>650 nm) and photoacoustic properties and are consequently preferred for in vivo applications. However, increasing the length of the polyene or conjugated rings may render the cyanines prone to photolysis. One design strategy involves the introduction of a rigid chlorocyclohexenyl ring into the methine chain for enhanced stability [71]. The most utilized dye of this family is indocyanine green (ICG), which is approved by the U.S. Food and Drug Administration (FDA) for use in humans [24]. It has been used in clinical trials for angiography, liver function testing, sentinel lymph node (SLN) mapping (mainly due to its ability to bind rapidly to plasma albumin), and to study the growth or/and treat tumors at the preclinical level [38,72]. In addition to its diagnostic utility, it can also act as a therapeutic agent due to its photothermal therapy properties, converting 88% of the amount of light absorbed into heat, and also in photodynamic therapy by producing reactive oxygen species after repeated radiation [73].

BODIPY (borondipyrromethane) dyes include the general structure of 4,4′-difluoro- 4-bora-3a,4a-diaza-s-indacene and exhibit notable fluorescence and photoacoustic properties [47]. These molecules possess thermal and photochemical stability and can be utilized in a wide range of applications, including imaging and photodynamic therapy as photosensitizers. Various forms of porphyrin have been used as photosensitizers [74], while dyes, such as Μethylene Βlue and Εvans Βlue, are typical examples used in diagnosis (and more rarely in treatment) of cancers such as cutaneous melanoma [7].

**Table 1 pharmaceutics-14-00362-t001:** Representative examples and the physical properties of chromophores with photoacoustic properties that have been used within cancer theranostics. The name of the chromophore, its chemical structure, its maximum wavelength of absorption (*λ*_ab_ (nm)) and emission (*λ*_fl_ (nm)), the molar extinction coefficient (ε, (M^−1^ cm^−1^)), and the fluorescence quantum yield (Φ) are presented.

Dye	Chemical Structure	*λ*_ab_ (nm)	*λ*_fl_ (nm)	*Ε*_max_ (M^−1^ cm^−1^)	Φ	Ref.
Protoporphyrin IX	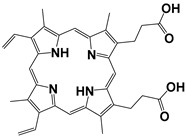	405	635	40,603	<0.01 (±0.01) in water	[75]
Methylene Blue	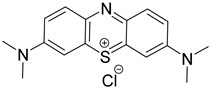	665	691	95,000	0.52 in ethanol	[76]
Indocyanine Green	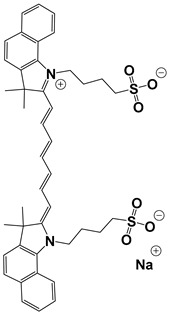	779 (in water) 815 (in blood)	805 (in water) 815 (in blood)	156,000 ± 6000	2.9 ± 0.2 (in water) 13 ± 1 (in blood)	[77]
PyBODIPY-7	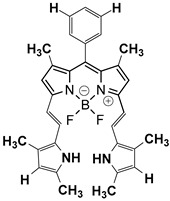	734	below the device detection limit	55,800	17	[78]

### 3.2. Selecting the Optimal Anticancer Drug during the Design of Cancer Theranostics with Photoacoustic Properties

Cytotoxic chemotherapy remains an integral part of cancer treatment and the National Cancer Institute (NCI) describes more than 600 FDA-approved anticancer drugs [11]. There is, therefore, no compelling argument to develop new cytotoxic drugs, as it represents a cost- and time-inefficient approach. As such, the repurposing (use in an application distinct from its primary indication) of agents is an appealing strategy. When designing a theranostic agent, the most appropriate anticancer drug can be selected from the large pool of available cytotoxic agents. This choice can be made based on various factors, including its mechanism of action and its activity against the target cancer cell type. Drugs are classified according to their mechanism of action, for example, antimetabolites (e.g., methotrexate, gemcitabine), alkylating agents (e.g., nitrogen mustards), topoisomerase inhibitors (e.g., camptothecin, doxorubicin), microtubule inhibitors (e.g., paclitaxel) [13], and phototherapeutics (e.g., porphyrin, ICG). In addition, the desired drug should be selected according to its reactive handles (e.g., –OH, –NH_2_, –COOH), so that its linkage with the utilized PAI agent is feasible. For this purpose, an additional linker/spacer can be inserted. Critically, the specific linkages should not perturb the cytotoxic effects of the drug or the properties of the PAI agent, and therefore, should be incorporated after rational design. The structures of the most commonly utilized anticancer drugs are presented in Figure 5. Additionally, all the aforementioned anticancer drugs, along with a short description and their main drawbacks, are summarized in Table 2.

Gemcitabine (dFdC or Gem) is an FDA-approved anticancer agent (antimetabolite) used in the treatment of many solid tumors including breast, ovarian, non-small cell lung, and pancreatic carcinomas. It is a nucleoside analog of deoxycytidine, in which two fluorine atoms have replaced the two hydrogen atoms in 2′ carbon, bestowing it with high toxicity (non-specific). Its metabolism involves cell internalization through nucleoside transporters and its sequential mono-, di-, and tri-phosphorylation by phosphorylating kinases. Its main mechanism of action is through the incorporation of its tri-phosphorylated form into the DNA strand during DNA polymerization, leading to the termination of the polymerization process (masked chain termination). In addition to its non-specific toxicity, two important disadvantages of gemcitabine include its bioconversion to the inactive metabolite 2′,2′-difluorodeoxyuridine (dFdU) after deamination by cytidine deaminase and the acquisition of resistance by cancer cells [18,79].

Doxorubicin (Dox) is a classic anticancer drug used in chemotherapy either as monotherapy or in combination with other drugs. Its mechanism of action is based on its ability to intercalate within paired DNA bases and induce DNA strand breaks. It also inhibits the enzyme topoisomerase II and induces apoptosis [80].

Camptothecin (CPT) is an inhibitor of topoisomerase I (Top1), an enzyme with a fundamental role in DNA replication. It binds to both the Top1 enzyme and intact DNA strands through hydrogen bonds, preventing the re-binding of the promoted DNA and the Top1 cleavage from DNA [81].

Nitrogen mustards (NM), such as mustine, which contain chloroethylamine functional groups, were the first agents used to treat cancer. They are classified as alkylating agents as their mechanism of action involves the formation of DNA adducts (alkylation). The binding takes place between guanine N-7 of DNA in an irreversible manner resulting in the formation of DNA inter-strand cross-links and apoptosis [82]. However, these agents generally suffer from low aqueous solubility and structural instability.

Paclitaxel (PTX) is a microtubule inhibitor. Specifically, it binds to beta-tubulin stabilizing the microtubule polymer and protecting it from disassembly. It is used in the management of multiple solid tumors including breast, ovarian, esophageal, and lung carcinomas. Its main drawbacks involve its low aqueous solubility and its structural instability [83,84].

Methotrexate (MTX) is an antimetabolite of the antifolate type used in the management of various neoplastic disorders including breast, head and neck, leukemia, lymphoma, lung, osteosarcoma, bladder, and trophoblastic neoplasms, and acts as folic acid antimetabolite [85].

Tamoxifen (TAM) is a selective estrogen receptor modulator (SERM) widely and predominantly used in both early and metastatic breast cancer [86,87]. Its main drawback is the low aqueous solubility.

**Table 2 pharmaceutics-14-00362-t002:** Summary of the anticancer drugs utilized within theranostic structures. Their mechanisms of action and the most important drawbacks are also presented.

Drug	Short Description	Main Mechanism of Action	Main Drawbacks	Ref.
Tamoxifen	Sold under the brand name Nolvadex. Selective estrogen receptor modulator (SERM) used mainly against breast cancer	Blockage of the estrogen receptor	Low aqueous solubility	[86,87]
Mustine	Sold under the brand name Mustargen. Belongs to the family of nitrogen mustards. It has been used mainly against lymphomas	Alkylation of DNA	Low aqueous solubility, chemical instability, high uncontrolled toxicity	[82]
Doxorubicin	Sold under the brand name Adriamycin. Belongs to the anthracycline and antitumor antibiotic family. A classic anti-cancer drug used in chemotherapy against multiple cancers including breast and bladder cancer, and Kaposi’s sarcoma	Intercalation between base pairs	High uncontrolled toxicity and resistance of cancer cells	[80]
Camptothecin	It is an alkaloid natural product isolated from *Camptotheca acuminata*. It has been used mainly against breast, ovarian, colon, and lung cancers	Inhibition of topoisomerase I	Low aqueous solubility, chemical instability, high uncontrolled toxicity	[81]
Paclitaxel	Sold under the brand name Taxol. It is a terpenoid natural product firstly isolated from *Pacific yew*. It is mainly used against ovarian, breast, pancreatic, and lung cancers	Hyperstabilization of microtubule polymer	Low aqueous solubility, chemical instability, high uncontrolled toxicity	[83,84]
Gemcitabine	Sold under the brand name Gemzar. It belongs to the family of antimetabolites due to its resemblance with cytidine. It is mainly used against lung, breast, pancreatic, and bladder cancers	Masked chain termination	Enzymatic instability, high uncontrolled toxicity, resistance of cancer cells	[18,79]
Methotrexate	It is used as an anti-cancer and anti-rheumatic drug. It belongs to the family of antimetabolites due to its resemblance with folic acid. It is mainly used against leukemia, breast, and lung cancers.	Inhibition of dihydrofolate reductase	Low aqueous solubility and deleterious side effects	[85]
Porphyrin	Heterocyclic macrocycle compound consisting of pyrroles. Heme is a complex of porphyrins with Fe^2+^. It is utilized as a drug in PDT	Photosensitizer (PDT)	Inadequate light absorption, non-selective for cancer and cutaneous photosensitivity	[88]
Indocyanine green	Cyanine-based dye approved by the FDA. Its sulfonyl groups endow it with water solubility. It is utilized as a drug in PTT	Photosensitizer (PTT)	Low photostability and non-selective for cancer	[89]

### 3.3. Selecting the Optimal Photosensitizer during the Design of Cancer Theranostics with Photoacoustic Properties

Phototherapeutic agents are often invoked to kill cancer cells. Specifically, phototherapy is an emerging platform for cancer therapy that uses radiation at a specific wavelength via a phototherapeutic agent to induce phototherapeutic cytotoxicity. It is considered an advanced treatment option compared to current oncological approaches. It has advantages of non-invasiveness, excellent spatiotemporal controllability, and minimal side effects [90]. According to the mechanism of action, phototherapy can be divided into PTT and PDT. PTT involves the increase of the temperature of a specific part of the human body above the normal value of 37 °C for a specific period. Temperature is a key parameter for the bioavailability of biological systems. Exploiting this knowledge allows the development of agents that convert radiant energy into thermal energy through vibratory stimuli with the ultimate consequence of increasing the temperature to a cytotoxic level (41–48 °C) causing protein denaturation, oxidative stress, and coagulative necrosis. The most important parameters of this process are the intensity and duration of the temperature rise, which should be within defined frameworks, to control the process [91]. PDT uses a photosensitizing agent (PS), such as porphyrins, chlorins, and phthalocyanines, to absorb the energy of light and direct it to a substrate that induces cytotoxicity. In the majority of the cases, this substrate is molecular oxygen for which the state of lower energy is a triple-state. Electrons can pass through an intersystem crossing from the excited substrate of PS to the ground triple state of oxygen, causing its excitation and resulting in the formation of oxygen in the singlet excited state (^1^O_2_*), which is cytotoxic [90]. The photosensitizer is necessary to function as an “energy carrier”, to absorb high energy radiation, and to transfer it to other species without damaging the tissue.

As described previously, when electromagnetic radiation interacts with a molecule it can absorb an amount of energy equal to the energy difference between the ground state and a higher excited state (grey lines in Figure 6). During the de-excitation, the molecule relaxes to lower vibrational levels via vibrational relaxation (curved black dotted line in Figure 6) or to low energy states of the same spin (straigth black dotted lines in Figure 6), and eventually relaxes to the lowest vibrational level of the lowest excited state S_1_. From this moment onwards it can follow different paths for further stimulation. Relaxation in which the multiplicity does not change (and a photon is produced), is called fluorescence (blue line in Figure 6). Alternatively, there is also relaxation to the ground state that does not lead to photon production, but the energy is transferred through heat to the surrounding environment (PTT), which can eventually also lead to the photoacoustic effect after expansion of the surrounding material. It is also possible for an excited chromophore to exit the singlet manifold through the intersystem crossing, a forbidden transition where the multiplicity changes from the singlet to a triplet state. Then, it can relax through the emission of a photon, which is called phosphorescence (presented with a red line in Figure 6), or it can interact with nearby triplet state molecules, such as oxygen (O_2_). Oxygen has a triplet spin in the ground state and after such an interaction it is converted to singlet oxygen (^1^O_2_*), a toxic form that is damaging the nearby biological systems (Figure 6). This is the principle of photodynamic therapy [92,93].

Phototherapy-based techniques have enormous potential in oncology, but, so far, they are limited to complementary techniques. Both fluorescence imaging and photoacoustic imaging can be easily combined with phototherapy given that in all three cases excitation with radiation is required. For a photoacoustic theranostic agent to be effective as a photothermal agent, it should have high photothermal conversion efficiency, which is determined by the fraction of absorption relative to the sum of the absorption and scattering of light [94]. It is clear that these two techniques share multiple similarities, and it would be useful if they were combined. Concerning PDT, the imaging agent frequently has photosensitizer properties (e.g., porphyrin) (Figure 5 and Table 2). Conjugates of small organic molecules (cyanine, porphyrin, BOD-IPY, phthalocyanine, croconaine) show high photothermal conversion efficiency and ease of modification, in order to obtain excellent biocompatibility and specific toxicity in selected areas [71,95].

### 3.4. Selecting the Optimal Linker during the Design of Cancer Theranostics with Photoacoustic Properties

When designing a theranostic agent, the role of the linker (the moiety that tethers the imaging agent to the drug and/or to the tumor-homing element) is of great importance. The selection of a sub-optimal linker unit could lead to the abolition of the efficacy of the final theranostic. Hence, the linker should be carefully selected so as not to disrupt the anticancer efficacy of the drug, the affinity tumor-homing element, or the photoacoustic and physicochemical properties of the imaging agent. Furthermore, stability in peripheral blood circulation is critical to avoid premature and non-specific release of the drug. There is an immense variety of linkers that can be classified based on their length, stability, release mechanism, functional groups, and hydrophilicity/hydrophobicity. According to their stability, functional groups, and release mechanism they are usually distinguished as two generic categories: non-cleavable and cleavable linkers (Figure 7).

The use of a non-cleavable linker can lead to a prolonged enzymatic and/or chemical stability of the final theranostic agent. Their structure, therefore, remains unaffected by hydrolases, pH changes, and radiation. Non-degradable conjugates usually include a thioether bond, ether bond, or a triazole ring (Figure 7a) [96,97].

Conversely, theranostic agents with cleavable bonds utilize biodegradable linkers that can be hydrolyzed by a specific chemical or enzymic stimulus. In fact, to enhance the specificity for the tumor microenvironment, linkers are designed to be cleaved/denatured by cancer biomarkers or unique elements of the tumor microenvironment. Such theranostics can selectively release the active drug within the tumor microenvironment [98]. Cleavable linkers can be subdivided into four categories (Figure 7b): degradable in the cytosol (disulfide); degradable in lysosomes by an enzyme (usually ester, amide, and carbamate); degradable in lysosomes by acidic conditions (hydrazone and oxime); and cleavable by external stimuli (e.g., temperature, magnetic field, ultrasound, and light). Specifically, in the cytoplasm of cancer cells, glutathione overexpression is characteristic and can cleave various chemical bonds, including disulfide, selenium/telluride, diselenide/ditelluride, ferrocenium, and metal-thiol binders [97]. Among the bonds decomposed by acidic conditions, the most popular is the hydrazone bond. It can be hydrolyzed to yield the relevant ketone and hydrazide in lysosomes and endosomes of tumor cells (pH range of 4.3–5.5). Hydrolytic enzymes can cleave random ester or amide bonds, while enzymes such as cathepsin B and uterine metalloproteinases recognize and cleave specific amino acid sequences (for example, Val-Cit dipeptide can be cleaved by cathepsin B and Ala-Ala-Asn tripeptide can be cleaved by the legumain) [96,97].

Another category that is often invoked during the development of theranostic agents involves the self-immolative linkers (Figure 7c). Such linkers comprise a trigger group coupled to a core molecule (e.g., 4-hydroxybenzyl alcohol). The trigger might respond to a specific stimulus (e.g., an enzyme), which then causes spontaneous intramolecular disassembly of the theranostic agent to finally release the active drug. The specific stimulus leads to the disintegration of the trigger group and the creation of an unstable intermediate, which, after a sequential transfer of electrons, releases the active drug. The fragmentation process can occur through 1,4-, 1,6-, or 1,8-elimination or cyclization. By utilizing specific features found in diseased tissues, such as lower extracellular pH, uneven reductive states (e.g., increased glutathione levels), and enzyme overexpression (e.g., Penicillin-G-Amidase, b-Glucuronidase, Cathepsin B), theranostic agents can be endowed for programmable drug release [99]. 

### 3.5. Targeted Theranostic Agents

A better understanding of the physiology and metabolism of cancer cells equips scientists with new and innovative drug delivery strategies. Potential targets comprise any molecules that are either abnormal or overexpressed in cancer cells compared to normal cells. As such, a targeted theranostic agent might include a tumor-homing element that usually interacts with cancer biomarkers and promotes accumulation of the conjugate in the tumor area (e.g., a molecule that interacts with a receptor overexpressed in cancer cells and leads to endocytosis). The theranostic agent can be directly coupled to such a tumor-homing element (e.g., small organic molecule, peptide, protein, antibody, aptamer) or they can both be incorporated into more complex structures like nanoparticles to exploit the passive targeting by the enhanced permeability and retention (EPR) effect, which leads to their accumulation in the tumor. Representative examples are folate analogs, which target the folate receptor (chemical molecules), cRGD peptide, which targets ανβ3-Integrin (peptides), and panitumumab, which targets HER1 and HER2 (antibodies) [47].

Most solid tumors show an increased need for oxygen and nutrients. Thus, they exhibit high levels of vascular permeability factors, such as bradykinin, nitric oxide (NO), peroxynitrite (ONOO-), and protein vascular permeability factor (VPF), leading to angiogenesis. Their rapid growth rate results in the formation of new tumor blood vessels with defective morphology and abnormal architecture compared to normal ones. This facilitates the permeation of various drug formulations (e.g., nanoparticles) to the tumor site [100]. Accumulation becomes even more effective if passive targeting is combined with specific targeting moieties, such as antibodies and specific molecules mentioned above (active targeting) [101,102].

The appropriate method of transporting the drug to the desired area plays a very important role and has sparked a huge research effort for the most effective targeted therapy, as well as the process of monitoring the course of drug delivery and the effectiveness of treatment. To achieve an efficient accumulation of the cytotoxic drug within the desired area and to avoid deleterious side effects, the aforementioned parameters have to be carefully considered, such as the microenvironment of the targeted area.

## 4. Representative Examples of Theranostic Agents Endowed with Photoacoustic Imaging Capability

There is a variety of ways to develop a theranostic agent with simultaneous therapeutic and photoacoustic properties. In most cases, the theranostic agent consists of a cytotoxic agent directly linked to an imaging agent (bimolecular theranostics: Drug-Imaging agent) or via different linkers, depending on the required/desired properties (trimolecular theranostics: Drug-Linker-Imaging agent) [103,104]. In certain cases, the incorporation of an anticancer drug is not mandatory as the imaging probe itself possesses therapeutic potency, usually via photothermal or photodynamic therapy (Single theranostic agent: Imaging agent that possesses therapeutic potential). The incorporation of an additional tumor-homing element to the aforementioned structures results in enhanced tumor accumulation and efficacy. Finally, most of the described theranostic agents can be formulated into more complex nanostructures that further increase the overall bioactivity and bioavailability profile.

### 4.1. Bimolecular or Trimolecular Theranostic Agents with Photoacoustic Properties

The first category of theranostics refers to the bimolecular and trimolecular theranostic agents. Such compounds consist of an anticancer drug tethered, either directly or via a specific linker, to an imaging agent that possesses PAI properties. 

To achieve a spatiotemporal dissociation and avoid unwanted side effects, there is a tendency to introduce drugs into the human body in the form of prodrugs, which remain stable and inactive in normal tissues, but are subjected to a chemical/enzymatic modification and, consequently, activation in the tumor microenvironment [18,105,106,107]. Accordingly, tethering an anticancer drug to a contrast agent via a cleavable linker, which can release the active ingredients in contact with a certain stimulus, is a commonly utilized approach. One of the most potent linkers is disulfide linkage [108,109,110,111,112], which is sensitive to reduction and can be degraded by high concentrations of glutathione (GSH), which is overexpressed in cancerous tumors [98]. This technique has been widely used to design prodrugs with fluorescent optical capabilities from chromophores, such as cyanine, semi-cyanine, and BODIPY [47], and drugs, such as gemcitabine and camptothecin [109,113,114]. 

A representative example is the coupling of the imaging agent Cy (IR780) to the anticancer drug methotrexate (MTX) via a disulfide bond, to yield the prodrug named Cy-SS-MTX. This theranostic agent can accumulate and release the active drug specifically in tumors after activation by GSH (Figure 8a) [115]. The dual-modal imaging prodrug absorbs at 654 nm and emits fluorescence at 750 nm, while, after interacting with GSH, both the fluorescence signal and the photoacoustic signal change (Figure 8b). Two wavelengths (680/808 nm) are used to excite the molecule. At 680 nm the prodrug provides a strong photoacoustic signal, which decreases after the interaction with GSH, while at 808 nm the signal from the prodrug is very weak, but increases after the interaction with GSH (Figure 8c,d). In vivo PAI experiments in tumor-bearing mice were performed at 0/0.5/1/4/8/24 h after intravenous injection of Cy-SS-MTX. At 680 nm, the signal was maximal 1 h post-injection (prodrug maximum concentration), while at 808 nm it was maximal at 8 h (complete disassociation of the prodrug) and was gradually diminished (Figure 8e).

Another example is the coupling of the anticancer drug paclitaxel with the dye methylene blue to give a photoacoustically-active conjugate (PTX-MB) (Figure 9a,b) [116]. The specific theranostic agent is activated after the oxidation of methylene blue, which in its reduced form does not provide any signal. The methylene blue core provides photoacoustic properties, while paclitaxel confers anticancer activity. The authors studied the antitumor activity in murine models for up to 8 h that showed the maximum increase in the photoacoustic signal (649% from 0–8 h) (Figure 9c). 

We would like to highlight that similar results to the aforementioned formulations could ideally be achieved by using any photoacoustic imaging probe (Figure 3) tethered with any anticancer drug (Figure 5) via the appropriate linkers (Figure 7). The selection is based on the required properties of the final theranostic agent (mechanism of action, photophysical properties, site of action, etc.). In most cases, the drugs are linked to the imaging agent via a covalent linker, as in the examples given in Figure 8 and Figure 9. In addition to disulfide bonds cleaved by reductive agents, such as glutathione, the literature describes a wide variety of cleavable linkages that are cleaved either by the acidic cancer microenvironment [117] or by specific enzymes. Jiang et al. and Zhang et al. designed two cyanide conjugates that bind through the same linker to gemcitabine so that their conjugates are inserted into the DNA strands [118,119]. In the first case, the meso position was reacted with gemcitabine and it was found to accumulate in brain tumors. In the second case, the molecule showed increased hydrophilicity and, thus, can be transported through the bloodstream and accumulate in cancer cells due to its inherent targeting capacity after the covalent attachment of serum albumin. We, therefore, can conclude that the coupling position, as well as the type of linker used, serve different roles and are of high importance [67]. 

Besides cleavable linkers that are typically utilized, in certain cases, non-cleavable linkers can also be used. For instance, Yang et al. developed a cyanine-based theranostic agent that contains tamoxifen. In this specific example, the linker consists of a 1,4-disubstituted triazole ring followed by an alkyl chain serving as spacer to fulfill molecular co-delivery requirements [120]. Another example by Yang et al. is a conjugate of a heptamethine cyanine dye—IR 78, tethered by an ether bond with the flavonoid-based natural product genistein [121]. The drug connects to the dye through the meso-Cl and a highly active hydroxyl group. This connection is necessary to improve the pharmacokinetic properties and the low potency limit of this natural drug. Indeed, this linkage improved these characteristics and showed increased accumulation in MCF-7 human breast cancer cells.

### 4.2. Thenanostic Agents Consisting of a Single Entity Endowed with Both Imaging and Cytotoxic Potential—Photothermal Therapy (PTT) 

This category refers to theranostic agents consisting of a single molecule that can be used for PTT. Two representative examples are those synthesized by Yan et al., who developed two organic phenazine-cyanine acceptor-donor-acceptor conjugates that absorb in the near-infrared region and exhibit both photoacoustic and photothermal capacities (Figure 10a) [122]. Their structure leads to redshift and absorption spectra of both molecules above 800 nm, rendering them capable of PAI and PTT. However, the authors, mainly aiming to enhance their poor water solubility, conjugated them to human serum albumin (HSA). The authors initially examined the photothermal capacity in vitro in PBS buffer, where they observed that the temperature could rise to ~50 °C which is able to kill cancer cells (Figure 10b,c). Then the photoacoustic capacity of the theranostic agents was assessed in vivo in nude mice. Nude mice were injected subcutaneously and then irradiated by an 808 nm pulse laser. The PA intensity was high in comparison with pure PBS buffer (Figure 10d,e). Finally, the capacity for photothermal therapy was examined in vivo, after intravenous injection in nude mice. The researchers observed that the temperature rose with time and reached ~43 °C after 10 min, which can damage cancer tissues. The thermal pictures are presented in Figure 10f.

Another approach involves the selective turn-on activation of the utilized PAI agent after stimulation by exogenous (e.g., radiation) or endogenous (e.g., pH change) stimuli. Meng et al. describe a structure of IR-822, a heptamethine cyanine dye that offers photoacoustic properties, conjugated to N1- (pyridin-4-ylmethyl) ethane-1,2-diamine (PY), which can accumulate effectively in tumors (Figure 11a) [123]. The linkage between the two components is stable and cannot be easily hydrolyzed. When the theranostic agent reaches its target (tumor microenvironment), it becomes protonated due to the slightly acidic pH and, through a rapid photoinduced electron transfer process, it is activated and as a result, increases its fluorescent and photoacoustic signals. Specifically, IR-PY was irradiated under a NIR laser of 690 nm, at different concentrations in PBS solution, and it was found that the PA signals increase linearly with increasing concentrations (Figure 11b). Furthermore, the ability of IR-PY to induce hyperthermia under NIR 808 nm laser irradiation was confirmed in PBS solution (Figure 11c). Finally, in vivo experiments for PAI were performed for 24 h in MCF-7 tumor-bearing nude mice (Figure 11d). The average PA intensity was calculated at specific time points after the injection of IR-PY (Figure 11e) and confirmed that IR-PY is a potential candidate for providing detailed information on tumor imaging [123].

At this point, it should be noted that in a recent publication, Lorenz et al. studied a structure-action relationship of heptamethine cyanine dyes, and after incorporating them into polymeric nanoparticle structures, the authors explored their properties. These scaffolds have photoacoustic and photothermal properties. In fact, the most effective were those containing a cyclohexane ring in the center of the dye and having certain characteristics, such as multiple rotating bonds and heavy atoms [124]. Heptamethine cyanine dyes have also been studied regarding their tumor-homing properties. Specifically, Mu et al. studied non-ionic heptamethine cyanine dyes, which can self-assemble into supermolecular discs. Thus, they are transported to the desired point by passive transport and, based on the characteristics of the dye (PAI and PTT), they can be cytotoxic [125]. Another class of pigments that exhibits photothermal properties is squaraines. Yao et al. managed to develop a strategy by which the resulting dyes have emission radiation in the range of 1000–1700 nm (NIR-II). Thus, they created an SQ1 nanoprobe with photoacoustic and photothermal properties [126].

Tian et al. recently developed NAD(P)H-triggered NIR probe, named Cy-N, designed to produce PA and NIRF signals, and also eliminate tumors via PTT (Figure 12a) [127]. The conception was based on the fact that cancer cells take up more glucose than normal cells, leading to the accumulation of NAD(P)H within hypoxic tumors, which consequently interacts with the reported NIR dye to exert its effect selectively in tumor sites. After evaluating the applicability of Cy-N in vitro, the authors explored the PA and PTT effect in tumor-bearing mice. Specifically, three different mice groups were used: Cy-N, Cy-N + glucose, and Cy-N + pyruvate. As shown in Figure 12b, PA signals are continuously increasing with time in the Cy-N group. In the other group, when the tumor is injected with glucose, a stronger PA signal over time is reported. However, in the other group that the mice are treated with pyruvate, the PA signal is reduced in tumor sites compared with the other two groups. The observed results clearly demonstrate that Cy-N is able to visualize NAD(P)H metabolism in tumors using the PAI modality. Afterward, the authors explored the efficiency of the PTT effect of Cy-N in tumor-bearing mice after irradiation at 600 nm (Figure 12c). In the mice injected with the control PBS, the temperature increased slightly from 28.7 °C to 33.4 °C, which corresponds to a poor PTT effect. When Cy-N was injected into the tumor-bearing mice, the temperature within the tumor area rose from 28.8 °C to 48.5 °C, displaying a more potent PTT effect. Notably, when the tumor-bearing mice were treated with Cy-N + glucose, the temperature within the tumor area rose from 28.8 °C to 66.0 °C, displaying a very efficient PTT effect. Finally, aiming to unveil the reduction of the tumor size with the proposed methodology, the tumors of these three different groups were dissected and measured. As shown in Figure 12d, the tumor of the Cy-N + glucose group is significantly smaller, unveiling the effect of the PTT derived from the proposed methodology.

Zhen et al. synthesized a multifunctional macrotheranostic agent with a trigger group specifically recognized by β-Galactosidase (βGal), an enzyme overexpressed in cancer cells and specifically in ovarian cancers (Figure 13a) [128]. Upon interaction with the enzyme, the theranostic agent is activated and can produce fluorescence and photoacoustic signals (Figure 13b). In the absence of the enzyme, the molecule shows an absorption maximum at 600 nm and does not produce optical signals, but when the enzyme is introduced, the CyOH molecule shows a new absorption peak at 688 nm, with a maximum intensity at 15 min. The PA signals of both probes at 680 nm significantly increase, with an increase in βGal concentration, but the signal of CyGal-P (the macrotheranostic) is 1.8-times higher than that of CyGal confirmed by both in vitro (Figure 13c) and in vivo experiments (Figure 13e,f). At the same time, part of the radiant energy is converted into heat causing hyperthermia (48 °C) (Figure 13d). The results were validated in SKOV3 xenografted tumor model (Figure 13g,h).

In addition to using different PAI agents and linkers, different trigger groups can be used, so the conjugate can be responsive to a specific stimulus overexpressed within solid tumors. Table 3 shows some typical examples of cyanines and semi-cyanines that possess PAI properties and are also decorated with certain trigger groups selective for various stimuli that are upregulated within the tumor microenvironment. These have been studied only regarding their diagnostic capability, but they could also be used for the development of theranostic agents in future works. 

IR806-PDA (Table 3) is a cyanine-based agent that is able to detect glutathione levels in living mice, and consequently produce fluorescence ratiometric signals in the NIR window (820 nm) avoiding a possible overlap with endogenous noise. IR806-PDA showed deep penetration and high selectivity for glutathione over other thiols, thus rendering it an appealing probe for the in vivo detection of GSH [129]. 

The second agent of Table 3, named HS-CyBz, consists of a cyanine dye core and can be used as an optical/PA dual-mode probe for in/ex vivo H_2_S imaging. HS-CyBz was found to accumulate in the liver of mice and showed great potential in detecting an H_2_S upregulation by showing enhanced emission intensity in the NIR region [130]. 

The third agent, named JW41 (Table 3), consists of heptamethine carbocyanine dye scaffold and is able to combine two different forms of imaging: Fluorescence and photoacoustic imaging. JW41 is designed to interact with the elevated levels of hydrogen peroxide (H_2_O_2_) in tumor sites via a boronic ester moiety. It also includes two 2-deoxyglucose moieties to enable selective tumor accumulation. The interaction of boronic ester with H_2_O_2_ triggers the enhancement of the fluorescence and PA signal, accompanied by a redshift of the spectra. In vivo experiments against subcutaneous MDA-MB-231 tumors in nude mice verified a targeted accumulation of JW41 within the tumor and demonstrated an effective non-invasive visualization of H_2_O_2_ [131]. 

The next agent, NR-NO_2_ (Table 3), includes a benzothiazole-xanthene core and is able to detect and image hepatic H_2_S elevation caused by metformin liver injury. The detection is performed in a dual-mode by both fluorescence and photoacoustic imaging. The specific signals increase when NR-NO_2_ interacts with H_2_S via the dinitrophenyl ether trigger, which is also used to quench the fluorescence prior to the recognition [132]. 

NR-azo agent (Table 3) is a xanthene-based NIR fluorophore with quenched fluorescence, tethered with a cytotoxic drug (chlorambucil-nitrogen mustard) via a hypoxia-labile azo linker. The complete molecule is encapsulated in liposomes and is designed to respond specifically to hypoxic tumors, releasing concurrently the NIR-emitting dye with high fluorescence intensity (710 nm) and the cytotoxic drug. Through this agent, the tumors can be visualized in a dual-mode by fluorescence and photoacoustic signals, and can also be treated by chlorambucil [133].

Drug-induced liver damage has concerned the scientific community, leading to the design of detectors that visualize such lesions. The following example of the DLP agent (Table 3) relates to such a detector using the multispectral optoacoustic tomography technique and fluorescence. This agent consists of a chromene-benzoindolium chromophore linked to a leucine aminopeptidase trigger via a 4-aminobenzylalcohol group. Upon interaction of the trigger group with leucine aminopeptidase (LAP), which is found upregulated in certain tumors, the photoacoustic and NIR fluorescence (733 nm) signals are increased indicating, either in vitro or in vivo, a possible elevation of the LAP levels [134]. 

CySO_3_CF_3_ (Table 3) consists of a water-soluble semi-cyanine core and a trifluoromethyl ketone moiety that is designed to interact with peroxynitrite (ONOO^–^) that is found in elevated levels in tumor sites. The NIR dye released upon the interaction with ONOO^–^ illustrates increased fluorescence and photoacoustic signals, rendering it an appealing diagnostic agent for ONOO^–^ detection even in living mice [135]. 

The last agent listed in Table 3, LET-3, is also a derivative of semi-cyanine. It includes a trigger group (phosphate group) that is designed to interact with alkaline phosphatase, which is present in diseased cells. LET-3 enabled the efficient visualization of ALP levels in tumor tissues through NIRF (730 nm) and PA imaging (710 nm). The high potential of the specific agent was validated both in in vitro and in vivo diagnostic experiments [136].

**Table 3 pharmaceutics-14-00362-t003:** Cyanine and semi-cyanine conjugates used for photoacoustic imaging with different trigger groups. Trigger groups are colored in blue; target groups are colored in green.

PAI Agents Decorated with Trigger Groups	Recognition Molecule	Reference
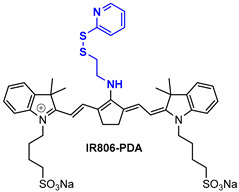	GSH	[129]
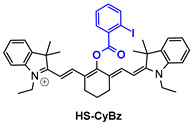	H_2_S	[130]
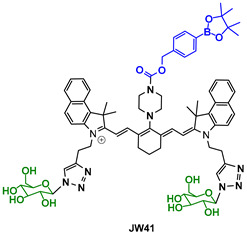	H_2_O_2_	[131]
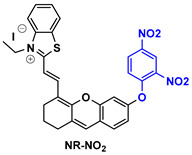	H_2_S	[132]
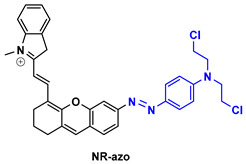	Hypoxia	[133]
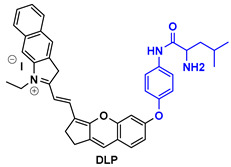	Aminopeptidase (LAP)	[134]
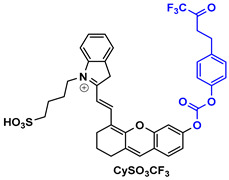	ONOO^-^	[135]
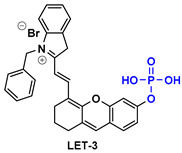	Alkaline phosphatase (ALP)	[136]

### 4.3. Thenanostic Agents Consisting of a Single Entity Endowed with Both Imaging and Cytotoxic Potential—Photodynamic Therapy (PDT)

Photodynamic therapy can be directly related to photoacoustic imaging. The main reason is that the stimulation of either the photosensitizer or the contrast agent is done by radiation. Photodynamic therapy is gaining increasing interest, particularly when the photosensitizer also possesses an imaging ability. Xu et al. examined the optical properties of the photosensitizer benzoporphyrin derivative monoacid ring-A (BPD), and its properties were compared with other commercially available photosensitizers. They proved that in addition to its photodynamic properties, it also possesses optical fluorescence and PAI capability [137]. Similarly, James et al. synthesized cyanine conjugates with a photosensitizer called 2-(1-Hexyloxyethyl)-2-devinyl pyropheophorbide-a (HPPH) via a linker coupling, which showed tumor-imaging capability, and in vitro and in vivo PDT efficacy [138]. Li et al. synthesized a monochromatic photosensitizer that its fluorescence was quenched until its activation by glutathione. The agent responds to this stimulus through a nitrovinyl moiety. Furthermore, it proved to be free of dark toxicity from the effects of certain atoms and non-specific phototoxicity, while after irradiation, it presented the desired cytotoxicity [139]. A representative example of a small-molecule theranostic probe was studied by Meng et al. [140]. This probe resulted from the conjugation of 5′-carboxyrhodamines (Rho) and heptamethine cyanine IR765 (Cy) via a disulfide bond, designed to be selectively cleaved by reductive thiols. Τhus, it possesses optical properties from both the rhodamine and the Cy dyes. Furthermore, it includes a pH-tunable amino group able to respond to pH changes (Figure 14a). This probe displays fluorescence and PAI signals (Figure 14b), while inducing reactive oxygen species for PDT. PAI was performed after the injection of RhoSSCy in subcutaneous MCF-7 tumor-bearing BALB/c nude mice (Figure 14d). The signal appeared in the first half-hour after the injection, and the maximum signal was observed at 15 h (Figure 14c). The accumulation capacity of RhoSSCy in the tumor area was studied by ex vivo experiments. The results showed that the probe accumulated mainly in the tumor and the liver, while there was a weak signal in the rest of the body. The control of photodynamic therapy was done through a fluorescence intensity technique, to visualize the production of reactive oxygen species (Figure 14e). The sample was irradiated with a 660 nm laser light at 30 mW/cm^2^ power, producing strong reactive oxygen species (ROS) levels and thus, proving to be a suitable photodynamic therapy agent (Figure 14f,g).

### 4.4. Theranostic Agents with Photoacoustic Properties and an Additional Tumor-Homing Element

Tumor-homing elements, such as peptides, are often invoked to bestow the theranostic agent with selective tumor localization efficacy. Capozza et al. described a diagnostic molecule after ICG coupling with a targeting peptide (Figure 15a) demonstrating that this approach is effective. It constitutes a typical example of peptide–drug conjugate for targeted malignant tumor diagnosis and possible therapy [38]. The cyclic tripeptide arginine-glycine-aspartic acid (cRGD), which is recognized by an integrin receptor (α_v_β_3_, overexpressed in cancer cells), leads to the accumulation of the imaging agent in the tumor [18]. It was used for photoacoustic imaging (Figure 15b), while it could also have therapeutic power, which would lie in the endogenous ability of ICG for photoconversion. For their studies the authors used the xenograft model (U-87MG-high α_v_β_3_ expression), which shows high vascularity and facilitates the distribution of the molecule in tumors (Figure 15c,d). This, combined with the ability of the peptidomimetic moiety, led to a strong accumulation in tumors. They also performed experiments on A431 cells that do not overexpress specific integrin receptors and the acoustic signal decreased much faster (Figure 15d). As such, the authors introduced a “proof-of-concept” for the development of new conjugates with photoacoustic and targeting abilities. It should be noted that the peptide could be replaced with any other targeting moiety depending on the individual targeted tumor site, such as the molecule designed by Bam et al., an antibody-based ICG conjugate that exhibits optical fluorescence capabilities and a photoacoustic signal [141].

Recently, Cheng et al. developed gold nanoparticle-based conjugates with Arg-Val-Arg-Arg (RVRR) peptides [142]. These peptides are specifically identified and cleaved by the furin enzyme. Therefore, the specific Au-RRVR decorated nanoparticles are designed to interact with furin, but also with acidic pH (5.5) in a synergistic manner. Both in vitro (in HCT-116 cancer cell) and in vivo (in HCT-116 tumor-bearing mice) experiments have shown that Au-RRVR NPs can accumulate and can be retained in the tumor area more efficiently than simple Au-COOH nanoparticles. It is important to mention that they possess PA and PTT properties, which were checked upon 808 nm laser irradiation (0.75 W/cm^2^) after IV injection. In fact, an increase in temperature of 31.7 °C was observed, leading to a significant number of dead cancer cells. Thus, the authors proved that they have developed an entrepreneurial therapeutic agent with increased tumor selectivity, which responds to an innovative imaging method (PAI) and phototherapy as a treatment.

### 4.5. Combination of Small Organic Theranostic Agents with Nanocarriers

Conversely, small organic molecules as theranostic agents possess certain disadvantages, such as poor water solubility, photobleaching, structural instability, and low photothermal efficiency [143]. To address these problems, the aforementioned organic molecules can be replaced by metals (e.g., palladium and gold) [144,145]. The metal particles show improved imaging capacity at higher wavelengths (NIR-I, NIR-II) and an enhanced capability in transforming the energy into heat. However, there are concerns about their non-specific toxicity and effects after long-term use [146]. Another classic approach that can be invoked to address the aforementioned problems is the encapsulation of the described organic molecules into complex formulations, such as nanocarriers. Hypotoxicity, biocompatibility, easy metabolism, the possibility of structural adaptability, and regulated spectrum led to the organization of organic dyes into such nanocarriers (liposomes, biodegradable polymeric nanoparticles, micelles, carbon nanotubes) with improved properties [36]. Among others, one of the most desirable and interesting properties of nanoparticles is the transition of the emission peak of the imaging agent to longer wavelengths [147]. This, in turn, leads to a greater penetration depth, which is translated in preclinical studies as the target of the most inaccessible organs. Furthermore, by coating the hydrophobic groups with either hydrophilic molecules or with suitable surfactant carriers, solubility problems in body fluids can be addressed. At the same time, one of the most important advantages of nanocarriers is the possibility of passive tumor targeting achieved by the EPR effect. 

A nanocarrier, besides the incorporation of organic compounds, is also characterized by an external surface whose properties can be appropriately modified to lead to improved ADME properties. The structure and architecture of each nanocarrier are also key parts of its design. Therefore, in addition to the aforementioned analysis of the utilized organic molecules, some typical examples of nanocarriers with different architectures that share common imaging and therapeutic properties are given below. The purpose of this work is not just to list the relevant literature, but also to create a total way of thinking about how to proceed from scratch with the synthesis of a small molecule-based theranostic, and what alternatives exist afterward (e.g., incorporation within a nanocarrier). The final assembly, derived by a careful design approach, could be an innovative and potent anticancer theranostic agent.

Nanoparticles (NPs) can be categorized according to their utilized components, such as carbon-based (fullerenes, carbon nanotubes, carbon nanodots) and inorganic nanoparticles based on metal oxides (e.g., iron oxide), metals (e.g., gold), and quantum dots (cadmium sulfide, cadmium selenide), according to Ju-Nam [148]. Metal nanoparticles are considered a milestone in the diagnosis and treatment of cancer, and among them, one of the most representative examples is the gold nanoparticles (Au-NPS) [149]. Au-NPS are often utilized as exogenous contrast agents with PAI properties due to their robust optical properties resulting from the surface plasmon resonance (SPR) effect [150]. Besides their desired optical properties even in the NIR-I and NIR-II windows, Au-NPS also show adequate biocompatibility and chemical inactivity under normal conditions, while they can become cytotoxic after irradiation by converting the energy of radiation into heat (PTT). The Au-based NPS usually invoked for such purposes include nanospheres and nanorod shapes. 

Au-NPS can act alone for both PAI and PTT purposes or can be used as drug carriers. Cheng et al. reported the development of photo cross-linkable Au nanoparticles consisting of photolabile diazirine moieties [151]. The specific light-triggered cross-linking elevated their SPR to NIR window and enhanced their PTT efficiency both in vitro and in vivo against 4T1 breast tumors. Wang et al. explored PAI in NIR-II window and PTT efficacy against ovarian cancer of a sandwich-type Au-NP coated reduced graphene oxide (rGO-AuNP), decorated with PEG on the Au-NP surface [152]. rGO-AuNP showed a promising profile against ovarian cancer by producing a strong NIR-II PA signal and, simultaneously, an important photothermal effect against ovarian cancer upon laser irradiation at 1061 nm. A representative example, where Au-NPs can be simultaneously utilized as drug carriers, was reported by You et al. [153]. The authors developed microspheres (1~15 µm) containing the anticancer drug PTX and hollow gold nanoparticles (~35 nm in diameter) with SPR in the NIR region. They achieved an efficient PTX release on-demand after irradiation with NIR light at 808 nm, resulting in important in vivo activity against U87 and MDA-MB-231 xenografted nude mice.

The nanoparticles can incorporate a diagnostic or therapeutic agent into their structure, by encapsulating it in the polymeric matrix or by adsorption (e.g., by π-π stacking or electrostatic interaction), or coupling (e.g., with covalent bonds) to its surface [154]. The imaging agent or therapeutic factor is usually encapsulated inside a hydrogel or a polymer (biodegradable or not). Hydrogel results from gelatinization (natural or chemical); the polymer results from polymerization (gradual growth or chain-growth polymerization, or photocatalytic); these materials create a closed system in the interior of which the encapsulation takes place chemically or physically. Zhang et al. used an amphiphilic DSPE-PEG 2000 co-polymer, and, by the nanoprecipitation method, they managed to encapsulate hydrophobic organic molecules (TI, TSI, and TSSI) [155]. They have successfully combined the following imaging methods: fluorescence imaging, PAI, and photothermal imaging. Fluorophores used are aggregation-induced emission (AIE) active. Therefore, they increase the fluorescence intensity upon the NIR-II region, as they accreted, restricting their motility and their bonds rotation. In addition to their excellent optical characteristics, these nanoparticles exhibit very important therapeutic properties in PTT and PDT synergistically. Specifically, after irradiation with a NIR laser (660 nm, 0.3 W cm^−2^) for 10 min, the temperature of 4T1 tumor-bearing BALB/c nude mice, after TSSI injection, increased from 37.3 to 54.8 °C. Simultaneously, irradiation at the right time was able to prevent the destruction of normal tissue by hyperthermia. Another representative example is that of Huang et al. who encapsulated indocyanine green in dendritic mesoporous organosilica nanospheres through a pore-composed structure. At the same time, they managed to encapsulate a catalase macromolecule that can decompose endogenous H_2_O_2_. The combination of catalase with the photoacoustic and photodynamic properties of ICG led to the development of a promising anti-cancer agent [17].

Liu et al. reported a one-component theranostic agent based on ICG in a J-aggregated state (with an average diameter of ~91.7 nm) (Figure 16a) [70]. This remains stable in a variety of solutions but denatures after cellular internalization. Τhis conjugation method increases the photothermal properties of ICG, while maintaining its optical properties. This study could be the engine for the development of more effective theranostics with increased biosecurity, high drug loading rate, high photothermal conversion efficiency, and ease of preparation [70]. PAI results showed more intense accumulation in the tumor area in the case of ICG J-aggregates, compared to the common ICG dye. However, the fluorescence intensity of the cells was similar in both cases. Authors believe that this can be explained by the negatively charged surface of these therapeutic agents, which limits their internalization by the cells. Liu et al. constructed a dual-modality nano-theranostic agent, using ICG as both diagnostic and therapeutic agent. They also utilized paclitaxel as a chemotherapeutic drug, leading to chemo/photothermal synergistic antitumor activity. PTT can be achieved via laser irradiation, while, simultaneously, the NPs can be treated with a near-infrared laser to rapidly release paclitaxel selectively at tumor sites [156]. A more recent example is that presented by Zuo et al. for photo- (by ICG) and chemo- (by DOX) therapy (Figure 16b). It is a carrier-free nano theranostic (size average from 219.9 nm to 183.8 nm) consisting of ICG as a photothermal agent and photosensitizer, and the mannose-thioketal-doxorubicin conjugate. Their surface is negatively charged and decorated with the carbohydrate mannose. Mannose interacts with lectin receptors, inducing an endocytosis mechanism. Then, the thioketal linker in the presence of ROS gets hydrolyzed and, consequently, the anticancer drug DOX is released. In addition, ICG is also used for inductive ROS production, resulting in highly efficient cytotoxic effects [157].

Micelles are an interesting mode of drug transport formulation, improving solubility and stability. They have a unique nanoscopic architecture, small size, and easy adaptation for good biocompatibility with the drug of choice. They can be formed from amphiphilic block copolymers and can also be subjected to modifications to improve their properties, resulting in their widespread use in drug and photosensitizer drug delivery systems [158]. An intriguing example is that published by Zhang et al., wherein the core contains a hydrophobic aza-BODIPY molecule and the hydrophobic paclitaxel (Figure 16c). In this study, Pluronic F127 was used as a surfactant, which converted it to micellar form (average diameter of ~18.5 nm) and endowed it with aqueous solubility and stability under physiological conditions. The final structure has fluorescence capacity in the NIR-II region and photoacoustic capacity, while at the same time it is used for chemotherapy (paclitaxel) and photothermal therapy (aza-BODIPY) [159]. 

Xiao et al. developed an aza-BODIPY photosensitizer decorated with a 2-pyridone functional group (Figure 16d) that bore photoacoustic imaging capabilities for targeted photothermal therapy, in addition to ^1^O_2_ production [160]. The ability of the molecule to increase the temperature in the solution was studied in vitro. In fact, there was a rapid increase in temperature from 28.8 °C to 49.2 °C within 6 min. This molecule could self-assemble with the amphiphilic polymer DSPE-mPEG_2000_ to form the surface of nanoparticles (with diameter 66.3 ± 6.6 nm), able to exploit the EPR phenomenon for tumor targeting, and demonstrating excellent tumor inhibition (93.4%). This was confirmed by studies in HeLa cells after being treated with different BDY NPs concentrations in the presence and absence of irradiation (660nm). The authors then performed studies in HeLa tumor-bearing nude mice at different time points and recorded the intensity of fluorescence and photoacoustic imaging. These results show that the photoacoustic signal is maximal at 6 h after injection. BDY NPs generate ^1^O_2_ and exhibit photothermal conversion efficiency of 35.7%. Even in the absence of light, the hyperthermia-promoted thermal cyclo-reversion from 2-pyridone endoperoxide leads to satisfactory production of ^1^O_2_. Through this conjugation, the authors propose photothermal synergistic sustainable phototherapy for satisfactory results in clinical settings.

Another organizational structure is semiconducting polymer nanoparticles (SPNs) that mainly improve the light and thermal stability of theranostic agents [161]. Sun et al. developed a zwitterionic polymer based on the organic dye perylene diimide, which was used for photoacoustic guided therapy via PTT and PDT. The novelty, in this case, is that the activation after NIR-light irradiation can occur under a single wavelength (600 nm) for both the optical and therapeutic properties of this polymer [162]. Such structures may also encapsulate a drug to provide synergistic therapeutic activity. Jiang et al. constructed a nanocarrier -DSPN- with a very simple and stable structure based on an amphiphilic polymer (PEG-PCB), which develops excellent hydrophobic and p-p interactions with the drug (doxorubicin), which is trapped inside the structure (diameter increased from 40 nm for PEG-PCB to 100 nm for DSPN). The hydrophilic moiety makes the vector stable in the plasma of living organisms. The amphiphilic backbone has both therapeutic and optical properties allowing effective chemo-photothermal therapy (Figure 16e) [163].

To further increase the residence time of the agent in the body by avoiding its rapid elimination, a small molecule with optical properties can be dispersed in a complex of long-circulating single-walled carbon nanotubes (SWCNT), such as those proposed by Xie et al. with the Evans Blue (EB) dye. The specific formulation also includes the photosensitizer, Chlorin e6 (Ce6), and albumin. Thus, it can be used for PTT and PDT visual and audio guidance [164]. The relationship of photoacoustic and ultrasound imaging has been extensively studied by Hester et al., who have shown that ultrasound (due to its low cost, real-time imaging, high degree of penetration, non-ionizing, and non-invasive action) can be combined with innovative forms of treatment, such as PDT with significant implications for optimizing anti-cancer therapy [165,166].

Liposomes are derived from a bilayer of phospholipids that form spherically enclosed colloidal vesicles, formed in an aquatic environment while part of it is enclosed inside. Xu et al. developed a nanoliposome where ICG and iodinated CTIA iohexol (Omnipaque^®^) were incorporated into its hydrophilic nucleus for X-ray CT imaging and induced PDT. ICG functions here both as a PTT and PDT factor, a phenomenon enhanced by the presence of the iodine heavy atom, which increases the PS triplet level, resulting in higher singlet oxygen yield [167].

A typical structural organization that effectively satisfies both ultrasound imaging and photoacoustic imaging is that of microbubbles. Microbubbles trap gas in their structure, and, due to their different sound conductivity from that of liquids and biological tissues, they can be used as effective contrast agents. Their signal intensity can be advanced by increasing the concentration in the desired area, either through passive (EPR effect) or active (with a cancer biomarker) targeting [168]. It is noteworthy that a drug or a photosensitizer can be incorporated into the surface of the microbubble or encapsulated within its nucleus. In this way, the microbubbles can function not only as contrast agents, but also as carriers of special molecules for the development of treatment techniques. Their surface can be modified (e.g., with lipids) to modify their properties, such as stability or tumor-homing capability [169]. Studies on the topic of microbubbles that combine therapy are limited for two possible reasons: (i) they show reduced drug loading capacity and (ii) reduced tumor accumulation capacity due to their large size, which does not allow them to pass into the tumor area, but, rather, remain in the bloodstream. A notable approach was developed by Yin et al. where indocyanine green-loaded microbubbles (average diameter from 500 nm to 150 nm) were used for simultaneous ultrasound-PA sequential imaging and multi-synergistic enhanced photothermal treatment (Figure 16f). Microbubbles’ surface was decorated with cyclic arginine-glycine-aspartic (cRGD) peptides, which provide them with a tumor-targeting angiogenesis ability. An ultrasound device with an amplitude of 500 kPa (LFUS500) was applied to convert the microbubbles into nanoparticles, triggering the EPR effect for even more selective ICG administration, and also increasing the photothermal conversion efficiency. Notably, conversion into nanoparticles occurred only within the tumor, under the guidance of real-time imaging. This led to a very efficient delivery in situ which was significantly superior to the conventional treatment plan [170]. PAI of the peptide-decorated microbubbles was tested at 800 nm and 860 nm and the targeting ability was tested in integrin α_ν_β_3_ positive and negative cells (U87MG and PC-3, respectively). PTT was also tested by an 808-nm NIR laser (1 W/cm^2^) for 5 min. In vivo experiments to evaluate the diagnostic and therapeutic capacity were performed on RM-1 tumor-bearing nude mice. In fact, after 3 min of irradiation, the temperature increased 5 °C, to 10 °C higher than the free ICG, and thus, the results showed a selective, biocompatible, and effective theranostic agent.

Liu et al. [171] showcased the theranostic potential of PEGylated CR760-based dye nanoparticles (average diameter of ~23 nm) (Figure 16g). Their surface was also decorated with c(RGDyC) peptides for more effective targeting ability, which was validated by confocal imaging in vitro in 4T1 breast cancer cells. They displayed efficient tumor targeting, while the high optoacoustic generation efficiency and photostability ensured strong optoacoustic contrast. These properties enabled phototherapeutic efficacy in vitro and in vivo, without recording any toxicity in major organs. Notably, the RGD-containing nanoparticles (CR760RGD-NPs) were administered intratumorally and then a laser beam was applied (5 min of irradiation with a 780 nm CW laser), leading to tumor elimination. By exhibiting a strong absorption peak at 760 nm, high photoacoustic generation and photothermal conversion efficiency, and photostability, the proposed nanoparticles appear to be very promising phototheranostic agents for combined cancer diagnosis and therapy.

**Figure 16 pharmaceutics-14-00362-f016:**
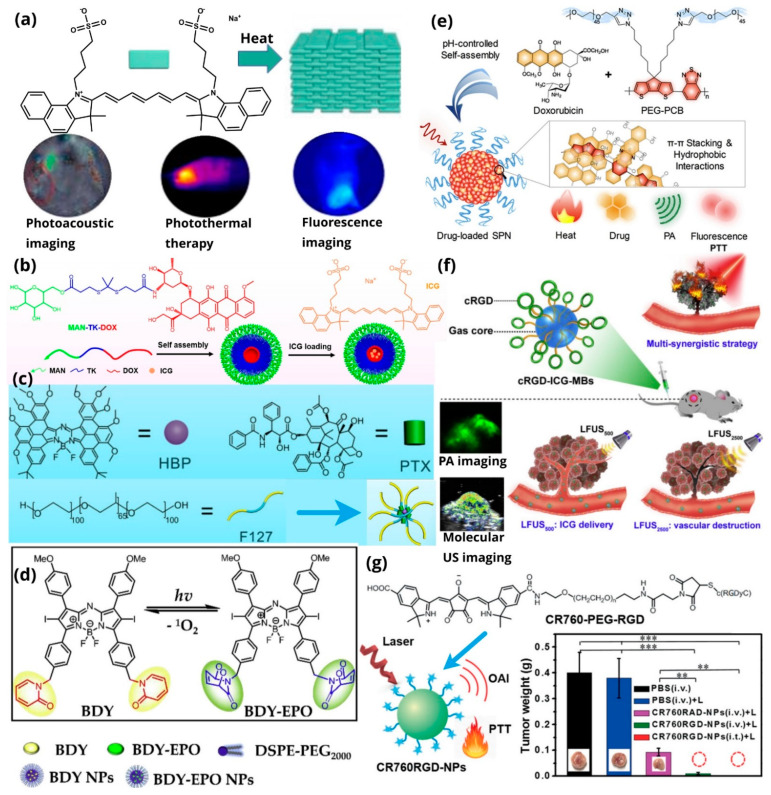
(**a**) Schematic illustration of the formation of ICG J-aggregate. Adapted with permission from [70], Nanotheranostics, 2017; (**b**) schematic illustration of ICG, light/ROS cascade-responsive tumor-recognizing nanotheranostics. Adapted with permission from [157], Elsevier, 2020; (**c**) schematic illustration of HBP/PTX micelles. Adapted with permission from [159], Elsevier, 2020; (**d**) schematic representation of the structural interconversion between BDY and BDY-EPO core. The path of the nanoparticle from the bloodstream to the organelles of the cell is also presented. Adapted with permission from [160], Elsevier, 2018; (**e**) schematic illustration of PEG-PCB and drug-loaded PEG-PCB. Adapted with permission from [163], Elsevier, 2017; (**f**) schematic illustration of ICG-loaded microbubbles. Adapted with permission from [170], Elsevier, 2020; (**g**) schematic illustration of CR760-PEG-RGD nanoformulation developed for PAI and PTT, and tumor weight after the administration of PBS, PBS (i.v.)+L, CR760RAD-NPs (i.v.)+L, CR760RGD-NPs(i.v.)+L, and CR760RGD-NPs(i.t.)+L. L = laser, i.v. = intravenous, i.t. = intratumoral. One-Way ANOVA with Tukey’s HSD test, ** *p* < 0.01, *** *p* < 0.001. Adapted with permission from [171], Wiley Online Library, 2021.

## 5. Clinical Translation

Photoacoustic imaging in preclinical research appears to be an emerging and promising hybrid technique for real-time molecular imaging and optically guided therapy. Notably, such efforts have also recently extended to a clinical level. Specifically, breast cancer, one of the most common cancers in females, has been used for translation from pre-clinical animal model research into clinical trials in humans [172]. The researchers take advantage not only of the fact that rapidly growing malignant tumors demonstrate neovascularization and faster extraction of oxygen from hemoglobin, but also the different photoacoustic properties of oxygenated and non-oxygenated hemoglobin [173]. Accordingly, using a device that creates images of US grayscale signals and combines color-coded photoacoustic features derived from different forms of hemoglobin, they achieved an efficient differentiation between benign and malignant breast tumors. High specificity and sensitivity were achieved towards this aim using a dual-mode US-PAI with non-ionizing radiation and without the use of an exogenous contrast agent. Such a test can have increased utility in the estimation of the size of the tumor and potentially reduce the number of false-positive tests and biopsies required for the accurate diagnosis of breast disease. The diagnostic utility of OA/US has been compared to standard diagnostic techniques in breast radiology (PET, scintimammography, color and power Doppler), and has proven to be promising. However, further studies are required to unveil its full potential in routine clinical practice [174]. In a similar study, the authors simulated real clinical conditions and they achieved the downgrading of benign masses using photoacoustic ultrasonography [175]. On the other hand, PTT for breast cancer has been tested in clinical trials for many years, also with promising results [176]. These outcomes have been discussed extensively by dos Santos et al. [177].

Τhe use of exogenous contrast agents has also been tested in clinical trials and the relevant paradigms are presented in Table 4. In particular, in pancreatic ductal adenocarcinoma (PDAC), a clinical study was performed with the drug Cetuximab-IRDye800 and is currently in phase 2. In this work, a near-infrared fluorescent and photoacoustic agent, as drug conjugate, was used to provide optical imaging and crucial information to the surgeon. The experimental agent accumulates efficiently in tumors via binding to the epidermal growth factor receptor and was tested for its ability to detect PDAC. The results showed higher mean fluorescence intensity in the tumor (0.09 ± 0.06) than in normal pancreatic tissue (0.02 ± 0.01) and a 3.7-fold higher photoacoustic signal in the tumor than in surrounding tissues [178,179].

One of the most discussed drugs is indocyanine green (ICG), which has been used in multiple clinical trials. In a relevant study, its ability to fluoresce in the NIR window is used for SLN mapping in colon cancer (Phase 2). ICG is initially injected next to the tumor, and its imaging capabilities can be used during laparoscopic surgery. This allows the removal of a small area (less than 5 cm) of the colon, with minimal side effects [180]. A further study in colorectal cancer (Phase 3) includes the testing of another common PAI agent, methylene blue. Oral administration in tablet formulation (MB-MMX) and follow-up monitoring through colonoscopy led to an absolute 8.5% increase in the rate of adenoma detection without mucosal contrast enhancement, compared with placebo [181]. Methylene blue has also been included in clinical studies of photodynamic therapy for other skin diseases [182].

Clinical trials have also been performed for therapeutic purposes through phototherapy, and the relevant paradigms are presented in Table 5. A typical example is in choroidal melanoma, utilizing the thermotherapeutic technique after drug injection of ICG-based Ranibizumab [183]. This study was completed in 2017, having reached phase 3. Photodynamic therapy is also in clinical trials with a wide range of photosensitizers. Classic examples of photosensitizers, such as HPPH and porfimer sodium, have been evaluated, leading to studies in stage I and II oral cavity squamous cell carcinoma and lung cancer, respectively, complementing traditional surgical procedures [184,185]. Both studies have reached phase 2. Other studies demonstrate that vitamin D pre-treatment (acute supplementation-neoadjuvant Vitamin D3) can enhance PDT for the treatment of actinic keratosis [186]. This study started in 2019 and is currently in phase 2. 

The diagnostic techniques of photoacoustic imaging and the therapeutic techniques of phototherapy have found applications in clinical studies separately, but a combination of these two, to our knowledge, has no clinical application until now.

## 6. Limitations and Future Perspective

Over the last years, PAI has shown great potential towards bioimaging applications, including the visualization of diseased organs, such as thyroid [187], liver [188], and skin [189], or the real-time monitoring of cancerous cells [190,191]. However, despite its exponential growth, its clinical translation is still limited due to certain factors, which are highlighted below along with proposed solutions. 

(1)The light sources (mainly lasers) utilized for the excitation of the PAI dyes. The currently utilized light sources possess restricted penetration depth and low pulse repetition frequency, while they also demand a high cost for their manufacture and maintenance. Additionally, their large dimensions provide an additional limitation in their usage. The construction of portable and economical light sources with fast-producing imaging pictures would provide an important impetus to the PAI field.(2)PAI systems are still slowly developed and they have not yet proved their potential effectiveness in humans. The scientific community should give additional attention to PAI approaches based on the appealing pre-clinical results and the widespread applications they can be utilized.(3)The acquired data need to be interpreted by experts in the field, thus, hindering their employment in hospitals. Therefore, novel algorithms and computational systems need to be constructed to facilitate the easy and rapid transformation of the complex data to interpretable information. Artificial intelligence is expected to play a major role towards this direction, as it will allow for the development of real-time visualization and classification software packages that will eventually increase the acceptance rate of the technology from the clinical personnel.(4)Even though it is widely acknowledged, there is still no consensus in PAI standardization traceable phantoms and procedures. This is an essential step towards the reproducibility of PAI data, the user-independent interpretation of the readouts, and the implementation of multicenter clinical trials, even with markedly different systems. Toward this end, it is required direct communication between the scientific community, industry, and the regulatory bodies, through the organization of international forums or consortia, such as the recently established International Photoacoustic Standardization Consortium (IPASC). The experience from the standardization in radiology can serve as a successful walkthrough and guide these efforts.(5)The nanoformulations for PAI purposes described within this review are also affected by the general concerns that arise regarding their safety in human health. Along these lines, nanotoxicology refers to the study of the toxicity of various nanomaterials. Novel nanocarriers with high biocompatibility need to be invented and utilized within clinical settings until they prove their effectiveness accompanied by minimal or zero side effects.

## 7. Conclusions

Innovative imaging techniques have established their importance in oncology, particularly in tumor diagnosis. Photoacoustic imaging (PAI) has attracted the interest of researchers over the last decade with significant progress. PAI has advantages, such as low cost, real-time non-invasive in vivo imaging, and can be easily combined with ultrasound and result in advanced images with combinatorial features, providing morphological, functional, and molecular data in a single diagnosis. 

Small organic molecules endowed with the ability to exert radiation of longer wavelengths (NIR-I and NIR-II region) and with increased radiation stability are continuously being developed. Interestingly, many of them can also act as anticancer drugs via photothermal or photodynamic therapy. Previous studies have documented that existing contrast agents can act as photoacoustic agents. Likewise, phototherapy has been extensively analyzed in studies examining photosensitizers and molecules that convert radiation into heat. 

In this review, we focused on theranostic agents that can be used for PAI and therapeutic (photothermal or photodynamic) applications. We primarily described the principles governing the development of such agents and extensively analyzed each component (drug, dye, targeting moiety, and linker) as they are of high importance and should be carefully sculpted. Additionally, we presented various representative examples of theranostic agents used for PAI and PDT, or PTT. The described examples also include complex formulations (e.g., nanoparticles) that offer several advantages including stability, water solubility, and tumor targeting. Finally, we reported some clinical studies that prove the ability of both photoacoustic imaging and phototherapy treatments in real applications.

We hope that this review will provide the impetus for the development of novel theranostic nanocarrier-based agents with photoacoustic properties. We believe that this approach will evolve into a potent tool for the accurate diagnosis and treatment of malignant tumors, and we envisage that it will be a standard option for future clinical applications.

## Figures and Tables

**Figure 1 pharmaceutics-14-00362-f001:**
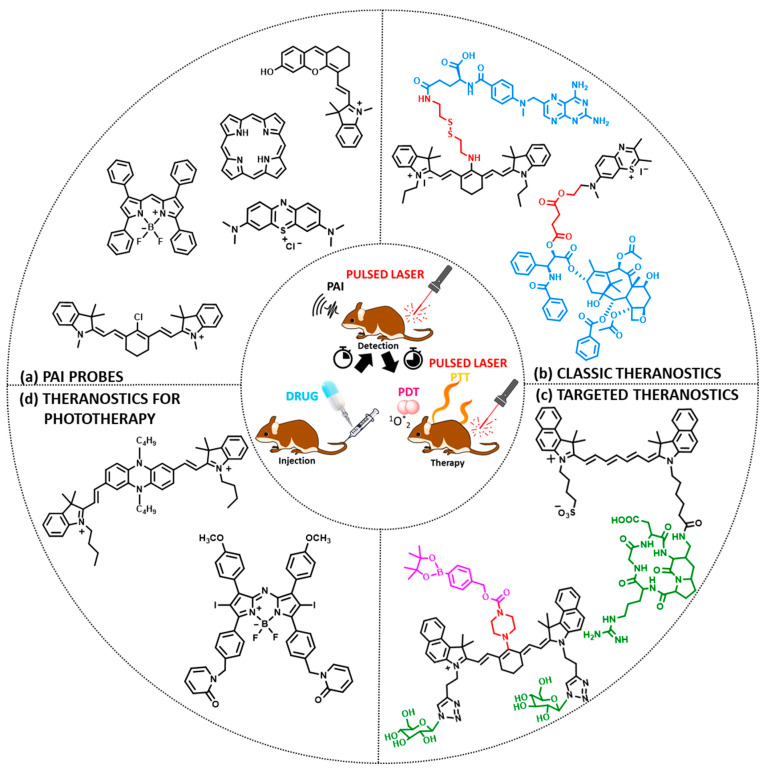
Representative examples of molecules utilized for photoacoustic purposes, that are described in this review. (**a**) Photoacoustic imaging (PAI) probes, (**b**) “classic” theranostics consisting of a drug and a dye tethered through various linkers, (**c**) targeted theranostic agents utilizing a tumor-homing element, and (**d**) theranostics utilized for simultaneous PAI and phototherapy. Photoacoustic agents are colored black, drugs are colored in light blue, targeting moieties are colored in green, the trigger group is colored in purple, and linkers are colored in red. PAI = Photo acoustic imaging; PTT = Photothermal therapy; PDT = Photodynamic therapy. Some of the illustrated components might have a dual role (e.g., indocyanine green dye analogs shown in black color in (**c**) offer both imaging and therapeutic properties).

**Figure 2 pharmaceutics-14-00362-f002:**
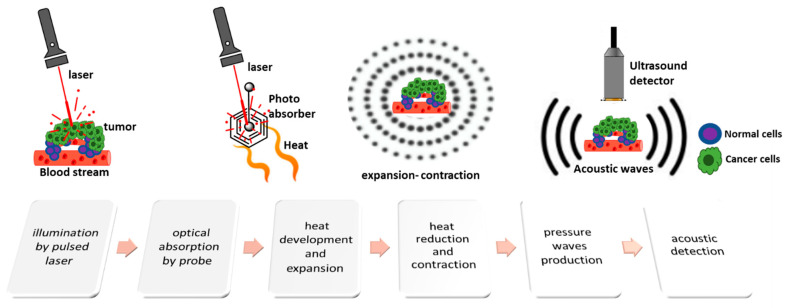
Schematic representation of the basic principles of photoacoustic imaging. The tissue with cancer cells (green cells) is irradiated with pulsed electromagnetic radiation, which is absorbed by a photo absorber, to produce sound waves that are detected by a focused ultrasonic transducer.

**Figure 3 pharmaceutics-14-00362-f003:**
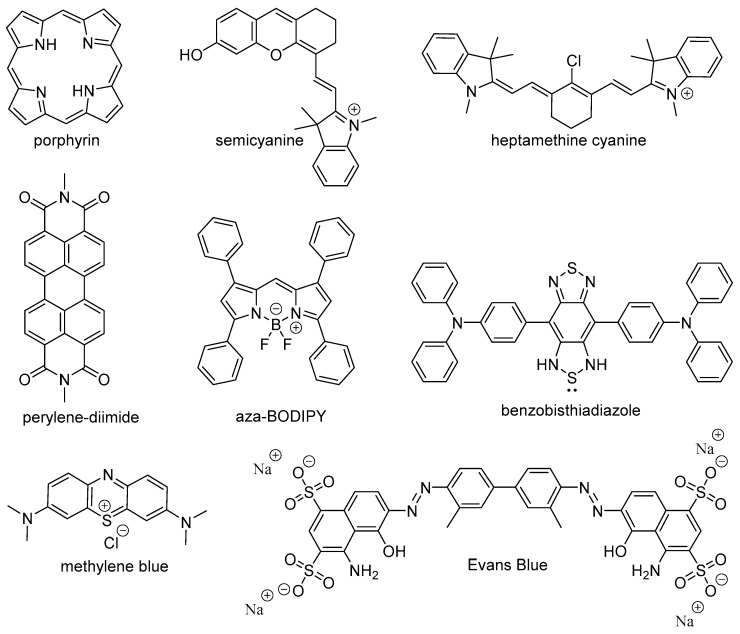
Basic chemical structures of organic-dye-based PA probes.

**Figure 4 pharmaceutics-14-00362-f004:**
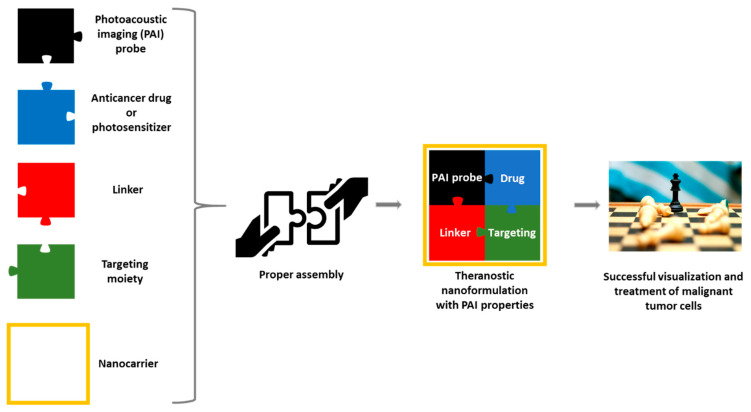
The generalized architecture to develop a theranostic agent with photoacoustic imaging properties, designed to visualize and treat malignant tumor cells. The different components of the “theranostic puzzle” can be combined in multiple, and different, ways to adduce the desired properties to the final fully assembled theranostic agent.

**Figure 5 pharmaceutics-14-00362-f005:**
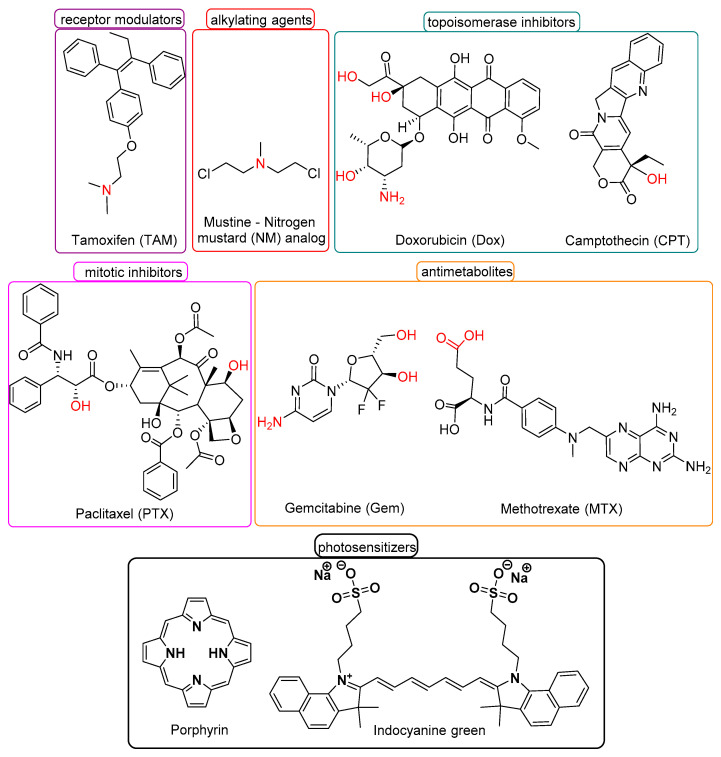
Structures of antitumor drugs and photosensitizers. Commonly utilized reactive groups are marked in red.

**Figure 6 pharmaceutics-14-00362-f006:**
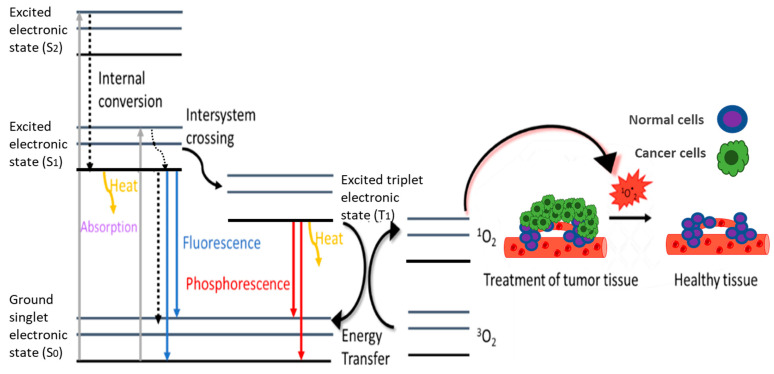
This figure presents a Jablonski diagram showing the basic processes that can take place when a molecule with absorption capacity is irradiated by a suitable wavelength of radiation. It also presents the basic principle of PDT and how it destroys the cancer cells of a tissue (green cells are destroyed).

**Figure 7 pharmaceutics-14-00362-f007:**
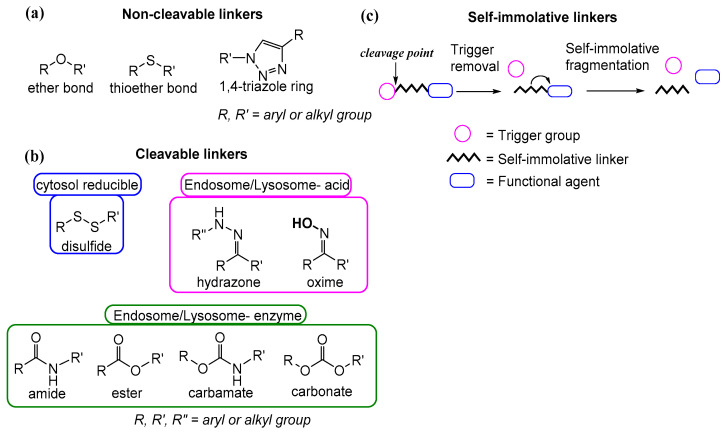
Categorization of linkers utilized in theranostic agents: (**a**) non-cleavable linkers, (**b**) cleavable linkers, (**c**) self-immolative linkers.

**Figure 8 pharmaceutics-14-00362-f008:**
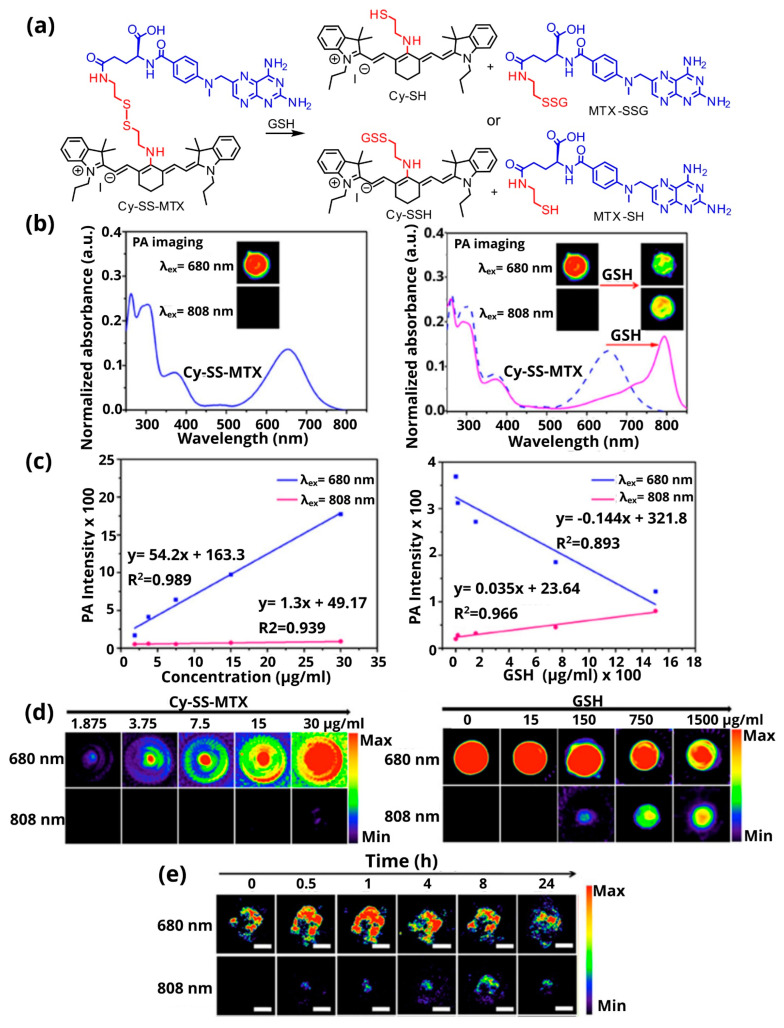
(**a**) Schematic representation of the mechanism of action of Cy-SS-MTX theranostic agent in the presence of glutathione (GSH); (**b**) presentation of the positive increase in photoacoustic signal of Cy-SS-MTX in the presence of GSH; (**c**) PA intensity of Cy-SS-MTX and responses of Cy-SS-MTX to GSH upon excitation at 680 and 808 nm, respectively, in varying concentrations; (**d**) the corresponding illustrations of PAI; and (**e**) in vivo photoacoustic imaging of tumor-bearing mice at 0/0.5/1/4/8/24 h after intravenous injection. Adapted from [115], Frontiers in Materials, 2018.

**Figure 9 pharmaceutics-14-00362-f009:**
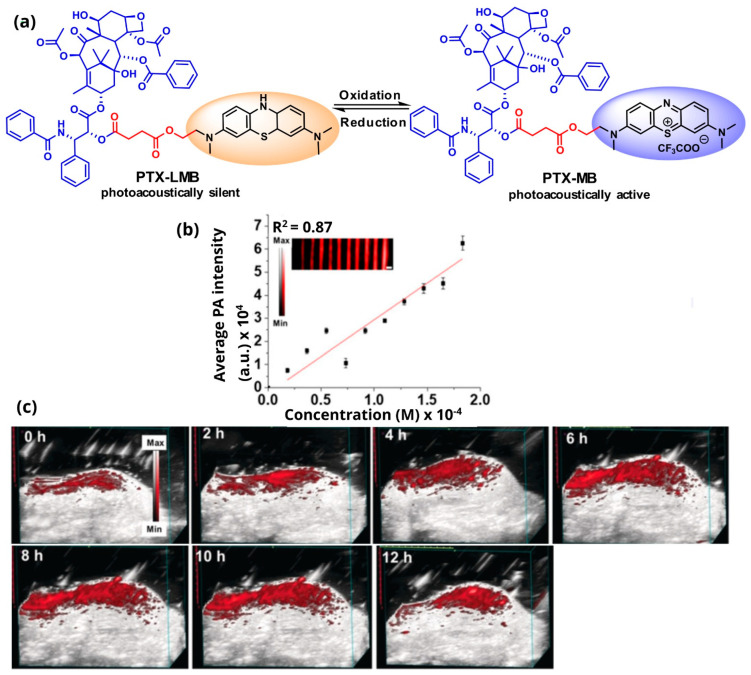
(**a**) Schematic diagrams of the mechanism of action of the PTX-MB conjugate; (**b**) presentation of the positive increase in the photoacoustic signal of the PTX-MB conjugate; (**c**) photoacoustic images after the subcutaneous injection of mice over time and the photoacoustic signal intensity increase at 0–8 h. Adapted with permission from [116], John Wiley and Sons, 2020.

**Figure 10 pharmaceutics-14-00362-f010:**
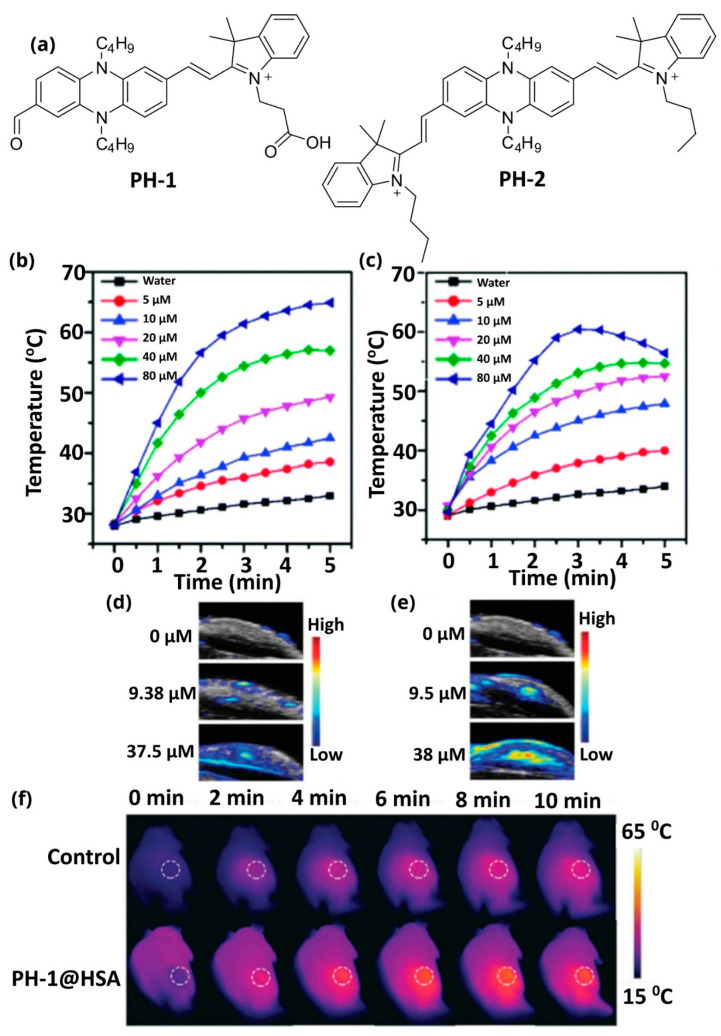
(**a**) Schematic diagrams of the PH-1 and PH-2 conjugates; (**b**) temperature curves of PH-1@HSA and (**c**) PH-2@HSA in different concentrations; (**d**) in vivo representation of the positive increase of the photoacoustic signal of PH-1@HSA and (**e**) PH-2@HSA in different concentrations; (**f**) thermal image of the nude mouse at different illuminating times. Adapted with permission from [122], Royal Society of Chemistry, 2018.

**Figure 11 pharmaceutics-14-00362-f011:**
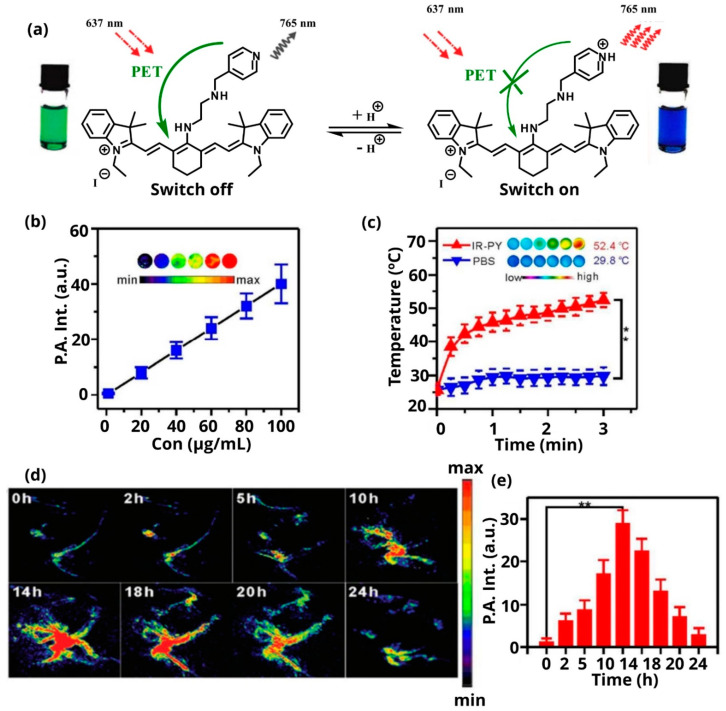
(**a**) Schematic diagrams of the IR-PY, pH sensing conjugates; (**b**) presentation of the positive increase of the photoacoustic signal of IR-PY in different concentrations; (**c**) photothermal temperature curves of IR-PY and control solution - PBS buffer- under continuous laser irradiation within 3 min; (**d**) PAI in vivo and (**e**) PA signal diagram intensity increment (** *p* < 0.01) of MCF-7 tumor-bearing living nude mice at indicated time points after in vivo injection of IR-PY. Adapted with permission from [123], Royal Society of Chemistry, 2017.

**Figure 12 pharmaceutics-14-00362-f012:**
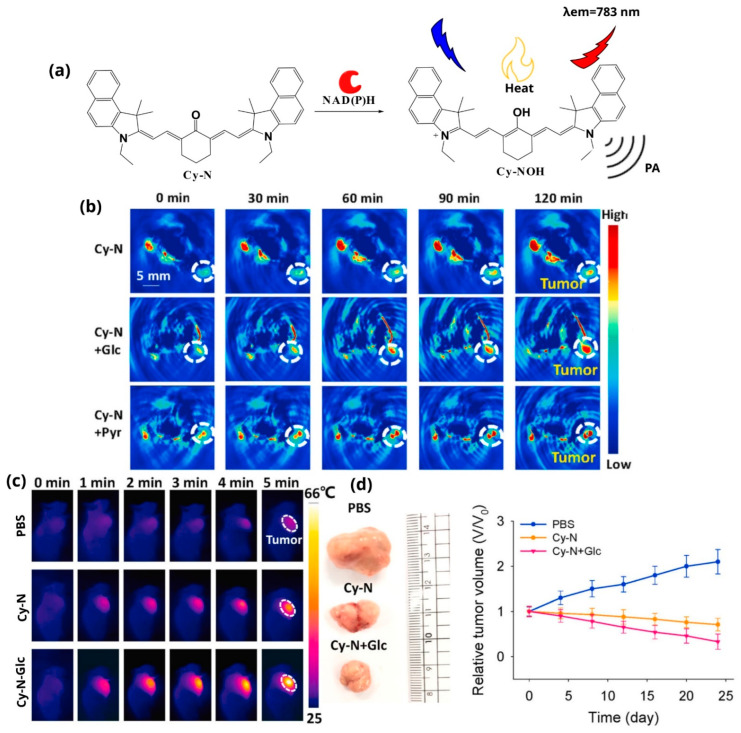
(**a**) Schematic illustration of the interaction mechanism of Cy-N with NAD(P)H to exert its effect; (**b**) the time-dependent PAI imaging of tumor-bearing mice; (**c**) PTT effect of the Cy-N probe against tumor-bearing mice after irradiation at 660 nm; (**d**) photographs of the excised tumors and tumor growth curves of the tumor-bearing mice using PTT and different treatments. Adapted with permission from [127], Elsevier, 2021.

**Figure 13 pharmaceutics-14-00362-f013:**
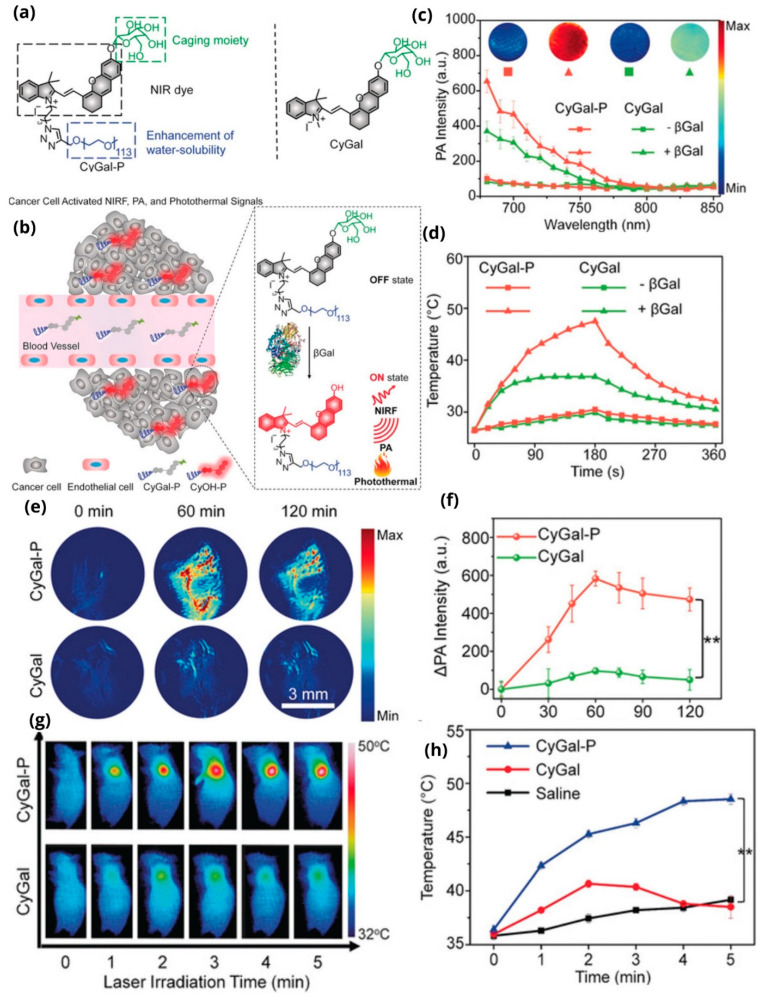
(**a**) Structures of the macrotheranostic probe (left) and the control probe without PEG (right); (**b**) schematic representation of the activation mechanism of the macrotheranostic probe; (**c**) the PA spectra of the probes in the presence or absence of βGal at 37 °C in vitro; (**d**) the temperature of the probes in the presence or absence of βGal after laser irradiation at 680 nm in vitro; (**e**) PA images of SKOV3 tumor-bearing living mice after in vivo injection of the probes; (**f**) PA intensity increment (ΔPA680 is denoted by a double star “**”) of CyGal-P and CyGal in SKOV3 tumor-bearing living mice; (**g**) IR thermal images in vivo after laser irradiation at 680 nm; and (**h**) diagrammatic representation of the photothermal phenomenon (Temperature intensity change is denoted by a double star “**”of CyGal-P and CyGal). Adapted with permission from [128], John Wiley and Sons, 2018.

**Figure 14 pharmaceutics-14-00362-f014:**
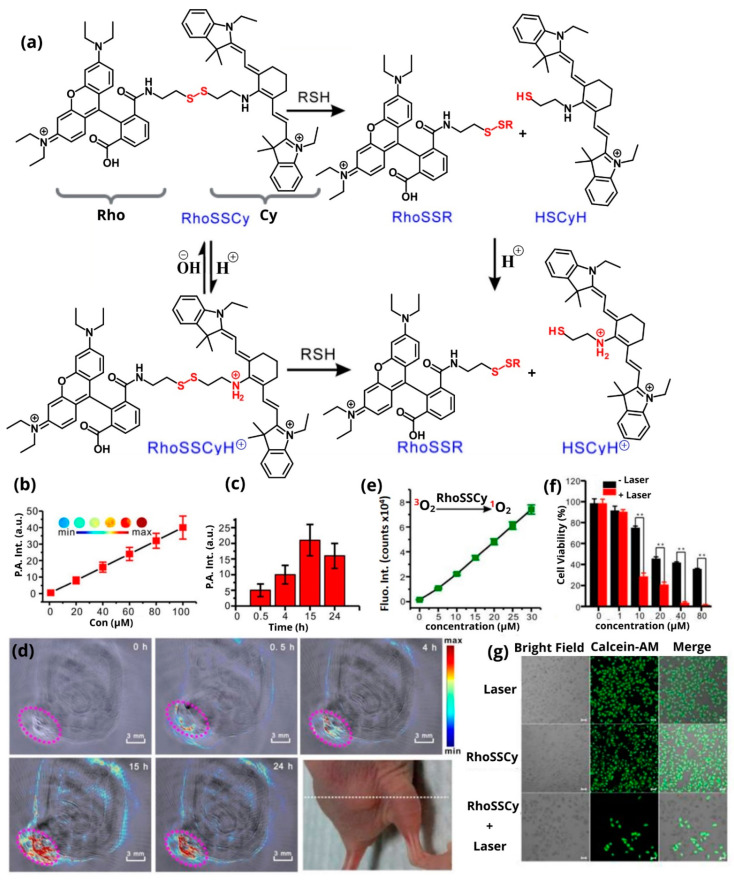
(**a**) Schematic representation of how RhoSSCy interacts with thiols and pH; (**b**) the correlation between PA intensity and RhoSSCy concentration; (**c**) graphic of PA signal as a function of time; (**d**) PA imaging of a tumor-bearing nude mouse, as well as real-time photos of the agent’s accumulation point; (**e**) graph showing the linear relationship from the release of ROS and RhoSSCy concentration; (**f**) images of MCF-7 cells visualized by fluorescence. Viable cells were stained green via calcein-AM dye. Bars, 25 μm (** *p* < 0.01); and (**g**) MCF-7 cell’s survivals using RhoSSCy in different concentrations after PDT. Adapted with permission from [140], Theranostics, 2017.

**Figure 15 pharmaceutics-14-00362-f015:**
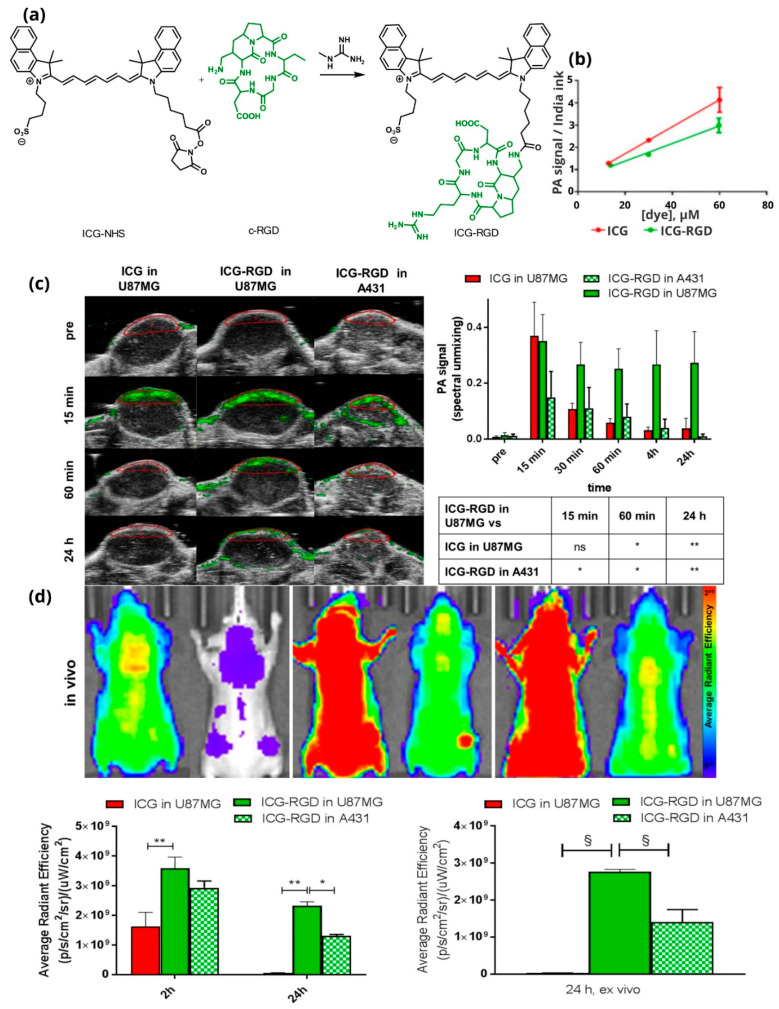
(**a**) Synthesis of ICG-RGD; (**b**) photoacoustic signal at different concentrations; (**c**) PA images of the tumor region acquired 15 minutes, 60 minutes, and 24 hours post-injection and at different time points after injection; (**d**) in vivo imaging in U-87MG and A431 tumor-bearing mice. Lower panels represent ex vivo optical images of the tumor (1) brain, (2) heart, (3) lung, (4) kidney, (5) liver, and (6) spleen, and (7) of A431- and U-87MG tumor-bearing mice injected with ICG and ICG-RGD. * *p* < 0.05, ** *p* < 0.01, 2-way ANOVA repeated measures followed by Bonferroni post-hoc test. Adapted with permission from [38], Elsevier, 2018.

**Table 4 pharmaceutics-14-00362-t004:** Exogenous contrast agents with PA properties in clinical trials.

Drug	Type of Cancer	Clinicaltrials.gov ID	Current Clinical Trials	References
Cetuximab-IRDye800	Pancreatic cancer	NCT02736578	Phase 2	[178]
Indocyanine green	Colon cancer	NCT01662752	Phase 2	[180]
Methylene blue	Colorectal cancer	NCT01694966	Phase 3	[181]

**Table 5 pharmaceutics-14-00362-t005:** Exogenous contrast agents with phototherapeutic properties in clinical trials.

Drug	Type of Disease	Clinicaltrials.gov ID	Current Clinical Trials	References
Ranibizumab injection and TTT-ICG based	Choroidal melanoma	NCT00680225	Phase 3	[183]
HPPH	Stage I & II oral cavity squamous cell carcinoma	NCT03090412	Phase 2	[184]
Porfimer sodium	Lung Cancer Metastatic Cancer	NCT00601848	Phase 2	[185]
Vitamin D_3_	Actinic keratosis	NCT04140292	Phase 2	[186]
Methylene blue/IPL	Verruca vulgaris	NCT04620785	Still not applicable	[182]

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
