# Peer review of "Design Principles Governing the Development of Theranostic Anticancer Agents and Their Nanoformulations with Photoacoustic Properties"

_pharmaceutics, 2022, doi:10.3390/pharmaceutics14020362_

Round 1

Reviewer 1 Report

The authors have focused on the design of novel theranostic anticancer agents and their nanoformulations with photoacoustic properties. The research contents are very interesting but there are some concerns must be addressed before being published in this journal.
1.     The authors should delete the Novel word from title
2.     In introduction part, author should introduce about Photothermal Therpay, as PTT has directly relationship with PAI
3.     Figure 1. PALSED Laser or Pulsed Laser?
4.     The author should check the abbreviations
5.     Table is required for 2.1 and 3.1
6.     The figures adopted from literatures need to discuss deeply, size and surface potential relationship with targeting ability with concluding statements.
7.     The authors should divide the 4.2 part (PAI+PTT/PDT) into PAI+PTT and PAI+PTT with some latest examples
8.     The authors should also pay more attention at Combination of small organic theranostic agents with nanocarriers, and provide detailed study about gold NPs as Au NPs are potential theranostic candidate (PAI+PTT),
9.     The authors should provide the problems, challenges and future prospective to get attraction of readers,
10.  The reference citation should be from past 5 years, please avoid old reference

Reviewer 2 Report

The review entitled “Design principles governing the development of novel theranostic anticancer agents and their nanoformulations with photoacoustic properties” offers a useful overview of the main strategies that allow the design of theranostic platforms for the simultaneous treatment and photoacoustic imaging of tumors.  After examining the key principles of photoacoustic imaging, as well as the properties of photoacoustic contrast agents, the manuscript provides a panoramic view on the most used “building blocks” of photoacoustic-based cancer theranostics, underlining their major properties and usages. Some representative examples of photoacoustic theranostic agents are also reported and accurately described and, finally, the authors focus on the clinical translation, exploring what kind of photoacoustic theranostic platforms showed major clinical relevance.

The themes covered are of big interest and, for this reason, the manuscript could be a useful guide for researchers who want to take their first steps within the field of photoacoustic-based theranostics. The basic principles listed in the first part of the review provide a good starting point for understanding the mechanics and potentials of the strategies successively described. The manuscript is also well written, providing an interesting and enjoyable reading.

Only a few minor comments that may improve the clarity of the manuscript are reported below.

  • In the Introduction section there is a large part of text without references (lines 43-57). Although many of the statements are well-known concepts, it is nonetheless suggested to add some references. In a similar manner, it could be better to also add references for the sentences in lines 229-231, lines 246-248 and lines 748-750.
  • The sentence “Phototherapy is based on the usage of different wavelengths of UV light” (line 63) is only partially correct and may be misleading. Given that phototherapy includes the use of contrast agents and platforms that can be activated not only by UV light, but also by other sources with wavelengths that fall into different spectral regions (such as VIS and NIR), the statement should be edited to also include those activation sources that are not part of the UV spectrum.
  • Lines 72-73 give a definition of a theranostic agent, underlining the importance of the employment of a spacer or linker to couple the imaging and the therapeutic agents. Even though the use of a linker is sometimes necessary and may give additional advantages to the theranostic system, it is not always mandatory. For example, nano-sized theranostic agents such as noble metal nanoparticles do not necessarily need a linker to exert their properties. Thus, it is suggested to edit the sentence in “A theranostic agent usually consists of an imaging agent coupled to a therapeutic agent through a spacer or a linker”.
  • In Figure 1 there is a little typo. In the central part of the figure, it is written “palsed laser” instead of “pulsed laser”.
  • In Figure 10 (f) rows labels are not clearly visible. It could be useful to adjust this panel part (maybe rescaling a bit). Similarly, in Figure 14 (d), column labels are not visible, making the image not clear for the reader. Please add the labels or, alternatively, add a more detailed description in the figure caption.
  • In lines 738-739, authors write about the use of “metal ions” to address certain disadvantages of small theranostic organic molecules. Taking into account the reported references, in which the use of pure metal nanostructures (oxidation state zero) is described, it should be proper to change the sentence writing “metals” instead of “metal ions”.
  • In the sentence in lines 768-772 could be useful to also cite carbon nano dots (CDs) among the theranostic carbon-based nanostructures and it is also suggested to remove “nanorods” considering that this is a classification based on the components and not on the shape.
  • The sentence in lines 795-797 is a bit confusing as it seems that BODIPY acts as surfactant. It is suggested to rewrite in order to make clearer the roles of each type of molecule.

Reviewer 3 Report

The current manuscript is comprehensive and well organized. I only have a few suggestions:

1, Please include a general description or findings of each study using the PAI agents listed in table 1. 

2, What are the current challenges of the PA theranostic agents for clinical translation/success? 

Reviewer 4 Report

The authors present a comprehensive literature review related to the development of novel theranostic anticancer agents with photoacoustic properties. This review is well structure and shows a progress from basic development/design of theranostic agents to their application in clinical trials. After reading the manuscript, I noticed a couple of misleading terms that are used interchangeably through the text. Additionally, there are a couple of sections that require some modifications and edits to assure that the main message is delivered. Lastly, the term nanoformulations in the title seems to be misleading since the description of principles for them is only described in one of the latest sections in this review. This can be my understanding of the term and there may be a difference on how it is being used by the authors. I will support this by the limited emphasis done in the introduction of this review about the nanoformulations. Below I present edits, suggestions, and modifications that should improve this article.

Line 19: “theranostic structures”, should this be theranostic agents?

Line 21: “incorporated into nanocarriers”, is it better to define this as formulated or designed as nanocarriers. This is based on the discussion presented in the text.

Line 47-64: there are drastic changes of topic within this paragraph. The authors go from diagnosis to therapy, then address advances in genomics to develop conjugates for active targeting. My recommendation is to make the transition between areas/topics smoother or to break this into multiple paragraphs.

Line 65: Either you are missing a section, i.e., section 3.3, or you have a typo. Section 3.3. does not correspond to PDT or PTT.

Line 66: first, the question is not formulated correctly. Second, consider starting this paragraph with something different than a question.

Line 79: Besides the fact that theranostic agents can be organized into more complex structures, as described in this review, the authors should consider other alternatives such as encapsulation within structures (microparticles and nanoparticles). These alternatives are commonly used for encapsulation and delivery of chemotherapeutic drugs with poor solubility. The concept of encapsulation is different than that of the organization into a structure.

Line 80: the authors should include other reasons of why a theranostic agent would be organized/encapsulated within a nanostructure.

Line 88: Can you please define or expand the concept of “classic” theranostic? This will be important for readers new to the topic or field.

Figure 1: Pulsed, rather than “palsed laser”

Lines 111-112: what is the relevance and impact of this sentence and reference to the discussion?

Lines 119-120: “Further improvement is still required” What does need improvement? What kind of improvements are you referring to? Is this related to fluorescence imaging?

Lines 160-162: This sentence needs work. For instance, redundant use of “could be further” and “assess possible their potential”

Lines 172-173: It is extremely confusing the way this sentence was written “the temperature and pressure do not rise above 0.1 K and 10 Pa”. Should you have said “the temperature and pressure increase are below 0.1 K and 10 Pa”?

Lines 174-175: What depths can be achieved? There is little to no reference in the text about the depths that can be obtained with photoacoustic imaging. Also, how real-time is this technique? I believe that some type of imaging processing must be done.

Line 189: Could you please explain the concept of “floor price”? is this a term normally used in the field? Or is it something that you want to introduce?

Figure 4: Are you suggesting that the ideal theranostic agent must have a photoacoustic imaging probe, a anticancer drug or photosensitizer, linker, targeting moiety, and nanocarrier? It is confusing if all these pieces must be present according to your experience. Additionally, a nanocarrier may not be needed if the other components/parts are suitable for biological applications. All in all, how do you justify your recommendation that the ideal theranostic agent must have 5 components?

Lines 321-355: I would suggest the authors to consider summarizing the different chemotherapeutic drugs and their properties, drawbacks, etc. with a table. The chemotherapeutic drugs are not part of the main discussion and as you mention there is no novelty in any of these drugs. You are only rephrasing descriptions that can be found in many articles.

Line 356: Should this be the space for Section 3.3? There is a huge change of topic that will benefit from a section break.

Lines 382-384: these sentences sound redundant. Please review to make sure that the main message is given.

Lines 382-397: this paragraph is hard to read and understand due to the different topics covered. Please consider using Figure 6 to describe some of the phenomena discussed in this paragraph.

Line 406: The term photothermal conversion efficiency (PCE) is introduced for the first time. The authors should describe and discuss it further since it is a relevant parameter for the design of theranostic agents with photoacoustic properties.

Lines 409-412: This sentence needs work

Line 420: “the linker should be carefully designed”. Would it be better to use “selected” rather than “designed”? Your recommendation is to use a library of well-known linkers rather than design one for the specific theranostic agent.

Lines 431-434: this sentence seems to be out of context. The discussion before it was related to linkers and then the authors switch to nanoparticles.

Line 470: The title of this section should be revised “Targeted chemotherapy based on theranostic agents”. What is the purpose of this section with the current title? Why are you focusing on chemotherapy? Should this be titled “targeted theranostic agents”?

Lines 485-488: Sentence needs work. Redundant use of factors and lack of structure.

Lines 575-577: This sentence is out of context

Lines 588-589: “tamoxifen (TAM), an anti-cancer …” this was already described in a previous section. No need to repeat information and extend the manuscript length.

Line 610: In the current context, “They” corresponds to the nude mice, is this correct?

Lines 610-614: Are these different experiments? It is hard to differentiate one description to the other.

Line 646: What do you mean by “macrotheranostic”? What is a macrotheranostic agent?

Lines 679-683: This a long sentence that requires some work.

Lines 688-690: Should this sentence/description be part of the nanocarrier section?

Line 691: Typo on HeLa cells

Line 692: Acronym for nanoparticles (NPs) has not been defined/introduced

Section 4.3: Are there additional references/examples of theranostic agents with a tumor-homing element?

Section 4.4: Please consider including the size of the nanocarriers presented for comparison

Lines 748-750: “one of the most … longer wavelengths”. Include a reference for this statement

Lines 753-757: Some of these ideas were already introduced and described in other sections.

Lines 768-772: These nanoparticle categories are misleading and incomplete. Please consider using more general categories or at least use them accordingly (e.g., nanorods are not a type of inorganic material, rather a morphology/shape)

Line 796: Please revise if aza-BODIPY molecule is used as “surfactant”.

Lines 807-809: Redundant use of structures

Line 820: “Photoacoustic imaging can be combined with ultrasound imaging”. This sentence is out of context

Line 830: redundant use of factor

Lines 864-866: This sentence needs work

Lines 887-888: “they could not only detect but also differentiate benign from malignant tumors”. This sentence needs work

Line 904: Use of the verb “reached”. The way this is written suggests that this drug only reached phase 2 and it did not move forward. Is this what the authors want to say? Or should you consider using a different term? For example, is currently in phase 2.

Tables 2 and 3: What is the purpose of these tables if the authors are not using nor referencing them.

Lines 992-993: “Finally, …” Please consider moving this sentence to the previous paragraph where methylene blue is discussed.

Table 3: What is the meaning of still not applicable?

Line 964: There is too much focus on nanocarrier-based agents. What is the authors definition of a nanocarrier? In my opinion, they are using the term interchangeably between drug, compound, and molecule of small size.

Other comments:

  • Revise the acronyms use in the article. They are introduced multiple times throughout the text
  • Revise how the references with last name (e.g., Xie, Lisi) are used and presented

Round 2

Reviewer 1 Report

The revised version of the manuscript is satisfactory